# Ion dynamics at the energy-deprived tripartite synapse

**Manu Kalia**[1]*, **Hil G. E. Meijer**[1], **Stephan A. van Gils**[1], **Michel J. A. M. van Putten**[2‡], **Christine R. Rose**[3‡]

**1** Applied Analysis, Department of Applied Mathematics, University of Twente, Enschede, The Netherlands, **2** Department of Clinical Neurophysiology, University of Twente, Enschede, The Netherlands, **3** Institute of Neurobiology, Heinrich Heine University Düsseldorf, Düsseldorf, Germany

‡These authors are joint senior authors on this work.
* m.kalia@utwente.nl

**Data Availability Statement:** The code for all the simulations performed are available at github.com/mkalia94/TripartiteSynapse.

**Funding:** This study was supported by the funds from the Deutsche Forschungsgemeinschaft (DFG), FOR2795 'Synapses under stress' to SAvG,

## Abstract

The anatomical and functional organization of neurons and astrocytes at 'tripartite synapses' is essential for reliable neurotransmission, which critically depends on ATP. In low energy conditions, synaptic transmission fails, accompanied by a breakdown of ion gradients, changes in membrane potentials and cell swelling. The resulting cellular damage and cell death are causal to the often devastating consequences of an ischemic stroke. The severity of ischemic damage depends on the age and the brain region in which a stroke occurs, but the reasons for this differential vulnerability are far from understood. In the present study, we address this question by developing a comprehensive biophysical model of a glutamatergic synapse to identify key determinants of synaptic failure during energy deprivation. Our model is based on fundamental biophysical principles, includes dynamics of the most relevant ions, i.e., $Na^+$, $K^+$, $Ca^{2+}$, $Cl^-$ and glutamate, and is calibrated with experimental data. It confirms the critical role of the $Na^+$/$K^+$-ATPase in maintaining ion gradients, membrane potentials and cell volumes. Our simulations demonstrate that the system exhibits two stable states, one physiological and one pathological. During energy deprivation, the physiological state may disappear, forcing a transit to the pathological state, which can be reverted when blocking voltage-gated $Na^+$ and $K^+$ channels. Our model predicts that the transition to the pathological state is favoured if the extracellular space fraction is small. A reduction in the extracellular space volume fraction, as, e.g. observed with ageing, will thus promote the brain's susceptibility to ischemic damage. Our work provides new insights into the brain's ability to recover from energy deprivation, with translational relevance for diagnosis and treatment of ischemic strokes.

## Author summary

The brain consumes energy to keep ion concentrations at normal working conditions. In the case of energy deprivation (ED), e.g., during a stroke, synaptic communication fails first. Inspired by our recent experimental work on ED, we formulated a novel computational model to explore initial events during ED. Our model reproduces time courses for

MJAMvP and CRR (Ro2327/13-1 and Ro2327/14-1). MK (project coordinators: SAvG, MJAMvP and CRR) was supported by project Ro2327/14-1. The funders had no role in study design, data collection and analysis, decision to publish, or preparation of the manuscript.

**Competing interests:** The authors have declared that no competing interests exist.

several ions from different experimental data. In some cases, the system returns to baseline upon restoring energy supply. In others, we observe that neurons and astrocytes cannot recover accompanied by cell swelling. There is a threshold depending on the depth and duration of ATP depletion differentiating these cases. Also, smaller extracellular spaces hamper recovery more. This result may explain clinical observations of increased vulnerability to stroke as the size of the extracellular space shrinks with ageing.

## Introduction

Information transfer at synapses [1] critically depends on the cellular availability of adenosine triphosphate (ATP), the main energy-carrying molecule in the body. Most of the energy consumption results from the activity of various ATP-dependent ion pumps, including the $Na^+$/$K^+$-ATPase (NKA). This pump transports $Na^+$ and $K^+$ ions to maintain physiological ion gradients across the cell membranes and various other ATPases involved in the release and vesicular reuptake of neurotransmitters like glutamate [2, 3].

Insufficient availability of ATP quickly results in synaptic transmission failure [2–6]. Depending on the depth and duration of energy deprivation (ED), this is accompanied by a loss of membrane potentials, cell swelling, and, ultimately, cell death [2, 5–7]. Several of these processes are well understood at a phenomenological level. For instance, if NKA activity is insufficient to counteract cellular $Na^+$ influx, the concentration of $Na^+$ increases within the neuronal and astrocyte compartments [6, 8, 9], and membrane potentials change. At glutamatergic synapses, the depolarization of presynaptic terminals causes the opening of voltage-gated $Ca^{2+}$ channels, resulting in $Ca^{2+}$ influx and subsequent glutamate exocytosis into the synaptic cleft. In addition, lack of ATP causes failure of plasma membrane $Ca^{2+}$-ATPases, while $Na^+$/$Ca^{2+}$- exchangers (NCX) may aggravate intracellular accumulation of $Ca^{2+}$ due to their reversal [8–10]. At the same time, the increase in neuronal $Na^+$ is accompanied by an increase in intracellular $Cl^-$ to maintain electroneutrality [11], resulting in cell swelling [12]. If NKA activity is completely absent such as in the core region of an ischemic stroke, ion gradients evolve to the Gibbs-Donnan potential [13]. All active transport fluxes have disappeared at this equilibrium, and the membrane potential equals the individual ions' Nernst potentials. This cascade of events is accompanied by failure of cellular glutamate uptake through excitatory amino acid transporters (EAATs), which are mainly expressed by astrocytes [14, 15]. The resulting toxic accumulation of glutamate in the extracellular space (ECS), coupled with intracellular $Ca^{2+}$ accumulation, triggers neuronal cell death via multiple pathways [16].

Despite these recent experimental advances in understanding the phenomenology of energy failure in the brain, there is only limited understanding of the interplay of the distinct transporters and ion fluxes of both neurons and surrounding astrocytes under these conditions. The tight anatomical and functional inter-relationship between pre- and postsynaptic compartments and surrounding astrocytes, established about 20 years ago as the tripartite synapse [1], adds to the challenging complexity of the biological system. The numerous dynamical and nonlinear interdependencies between different transport processes in the cellular compartments involved can eventually only be elucidated with biophysical models, calibrated with experimental data [17]. Motivated by these considerations, various biophysical models of the tripartite synapse have been introduced [18–20]. Some models focus on glutamate release and uptake in the extracellular space and $Ca^{2+}$ release from $IP_3$-sensitive $Ca^{2+}$ stores in astrocytes [21–23]. Other works have addressed mechanisms of gliotransmission via vesicular release [24, 25].

A few models describe neuron-glia interactions in pathological conditions, e.g., to identify mechanisms involved in cell swelling or spreading depression [11, 26–29]. Several key questions, however, remain. For example, the temporal aspects remain mostly unexplored, whereas experimental data shows that the duration of ED is a crucial parameter. Short durations allow neurons to recover [8]. Longer, even infinite, durations of ATP depletion lead to the Gibbs-Donnan potential [11].

A clinically highly significant finding is that the vulnerability to ischemia differs between brain regions. Here, the differential sensitivity of the cellular and biophysical determinants is not evident [30–32]. The age-dependence of the brain's cellular vulnerability to ischemic conditions is another highly relevant phenomenon that is not understood in detail [33]. The expression of many ion transporters changes significantly during postnatal development and maturation of neuronal networks. Ageing is associated with down-regulation of NKA and weakening pump activity [34] and is accompanied by structural changes such as the fine morphology of astrocytes [35]. However, it is unclear to what extent this contributes to the age-dependent sensitivity to ischemia. In addition, the relative size of the ECS changes significantly during life. Whereas the ECS volume fraction is about 20% of total tissue volume in adult rodents' forebrain, it is around 40% in neonates and shrinks to about 15% in aged animals [36]. The consequences of these changes for the functional interplay between the different compartments of the tripartite synapse and their relevance for the vulnerability of tripartite synapses to energy depletion are not understood.

Taken together, we hypothesize that there are a critical duration and depth of ATP depletion to induce a pathological state. This duration and depth depend on ion kinetics and geometry. To address this question, we study how the bulk ion species, i.e., $Na^+$, $K^+$, and $Cl^-$, react to switching off NKA for a specific duration and depth. Our model also includes $Ca^{2+}$ and glutamate. Although we report on their dynamics, we do not study these in detail as they do not represent the bulk of ions. Nevertheless, it is necessary to include both as they mitigate large $Na^+$- and $K^+$-currents during energy depletion.

We propose a novel model combining elements from existing models into a comprehensive, biologically realistic description. The model includes a neuron and an astrocyte within a finite extracellular space. For the active transporters and glutamate release, we formulate a correct electrodiffusive, ion conserving model. All ion fluxes included in this work have been studied and validated in simpler setups. For most models, it suffices to model $K^+$. For instance, Kager et al. [37] proposed a microdomain model accounting for all fluxes with glial behaviour modelled with a $K^+$ bath to study spreading depression. Later, other researchers [11, 38–41] have proposed electrodiffusively correct models. Building on these submodels, we use parameter values from the literature where possible. We calibrate the remaining ones using recent experimental data [8].

In this study, we first briefly describe the layout of our biophysical model of the synapse under normal conditions. We show that including the astrocyte is essential for preserving synaptic transmission. Next, we simulate energy depletion by switching off NKA. Depending on the duration and depth of ATP depletion, the system's state either recovers or ends up in a pathological equilibrium. We explain these results using bifurcation analysis, allowing us to identify critical factors determining vulnerability, including the size of the extracellular volume and NKA pump strength. Finally, our simulation results agree with experimental data regarding recovery [32], enabling us to explore how to promote the recovery to the physiological state. Our results aid in our understanding of the early events following energy depletion. Most notably, our simulations shed new light on the cellular basis of differential vulnerability of neurons to ischemic damage.

## A model for ion homeostasis at the tripartite synapse

We introduce a novel biophysical model, extending our previous work on a single cell [11] by incorporating more biological detail. The model in [11] contained a single neuronal compartment in an infinite extracellular space, whose volume is regulated by changes in osmolarity. Here, we model a neuronal and astrocytic soma, a presynaptic terminal, the perisynaptic astrocyte process, the synaptic cleft and a global extracellular space (ECS), illustrated in Fig 1. The volumes of the somatic compartments and ECS can change; the cleft, the presynaptic terminal and the perisynaptic astrocyte process have fixed volumes. In each of these compartments, we describe the energy-dependent dynamics of the ions $Na^+$, $K^+$, $Cl^-$, $Ca^{2+}$ and glutamate, including vesicle recycling, by ordinary differential equations (ODEs). We further assume that $Ca^{2+}$ and glutamate are confined to the presynaptic terminal, the perisynaptic astrocyte process and cleft. The concentrations of $Na^+$, $K^+$ and $Cl^-$ are assumed to be the same in the synaptic as well as in the somatic compartments. This allows us to work with both small molar changes of $Ca^{2+}$ and glutamate and large $Na^+$, $K^+$ and $Cl^-$ gradients across the somatic membranes, within the same compartmental framework.

## Model equations

For an ion $X$ in compartment $i$, we describe the dynamics of the number of moles $N_X^i$ and volumes $W_i$ by introducing ion channels and cotransporters $Y$. Their currents and fluxes $I$ are given by $I_Y^{X,i}$. The currents/fluxes $I$ may depend on gating variables $q$ that describe the non-linear opening and closing of the channels/cotransporters. We define concentration $[X]_i$ of ion $X$ in compartment $i$ by $[X]_i = N_X^i/W_i$. A summary of the notation used is shown in Table 1. Thus we obtain the following system of differential equations that describe the dynamics of $N_X^i$

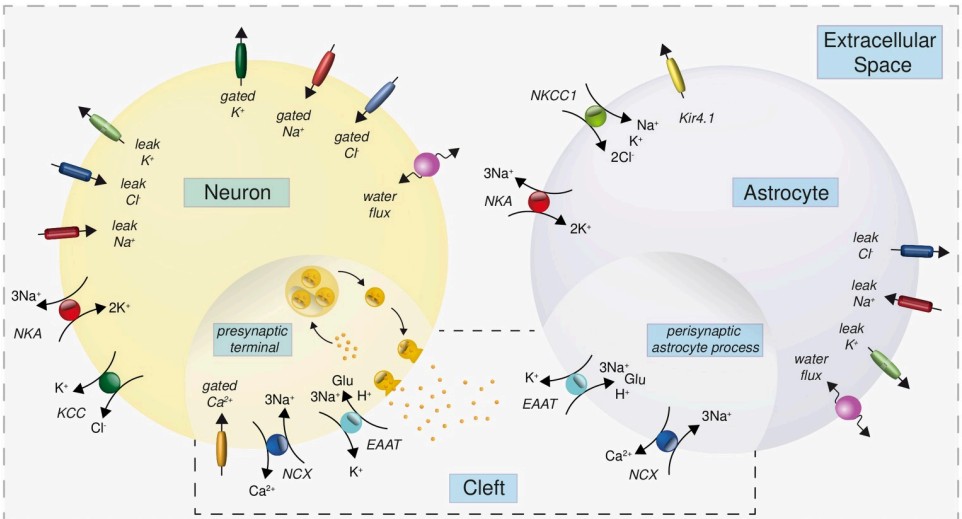

**Fig 1. Compartments, ion channels and transporters included in the modelling of the glutamatergic synapse.**
Shown are the three main components representing a presynaptic neuron, an astrocyte and the extracellular space (ECS). Each of these compartments also contains a synaptic compartment as indicated by the different shading and the additional box (presynaptic terminal, perisynaptic process and synaptic cleft, respectively). The largest ATP consumption in the presynaptic neuron and the astrocyte is by the $Na^+$/$K^+$-ATPase (NKA). At the presynaptic terminal, ATP is also needed to energize glutamate uptake into vesicles. The key transporters at the cleft are the $Na^+$/$Ca^{2+}$-exchanger (NCX) and the Excitatory Acid Amino Transporter (EAAT). NKCC1: $Na^+$-$K^+$-$Cl^-$-cotransporter. KCC: $K^+$-$Cl^-$-cotransporter. Kir4.1: $K^+$ inward rectifier channel 4.1.

**Table 1. Notation used in the model equations.**

| Notation | Description |
|---|---|
| $N_X^i$ | Molar amount of ion $X$ in compartment $i$. |
| $[X]_i$ | Concentration of ion $X$ in compartment $i$. |
| $W_i$ | Volume of compartment $i$. |
| $I_Y^{X,i}$ | Current/flux contribution of ion channel/transporter $Y$ with respect to ion $X$ in compartment $i$. |
| $V_i$ | Membrane potential with respect to compartment $i$. |
| $z_X$ | Valence of ion $X$. |
| $P_X^{X,i}$ | Permeability/strength/conductance of ion channel/transporter $Y$ with respect to ion $X$ in compartment $i$. |
| **Choices for i, X, Y** | |
| $i$ | $n$ (neuronal soma), $a$ (astrocyte soma), $e$ (extracellular space), $ps$ (presynaptic terminal), $pap$ (astrocyte process) or $c$ (cleft). |
| $X$ | Na$^+$, K$^+$, Cl$^-$, Ca$^{2+}$ or glutamate. |
| $Y$ | EAAT, NCX, NKA, KCl, NKCC1, Kir, G (Gated) or L (Leak). |

and $W_i$

$$\begin{cases} \dfrac{d}{dt}N_X^i = -\dfrac{1}{z_X F}\sum_j I_j^{X,i}, \\[2ex] \dfrac{d}{dt}W_i = \lambda_i \sum_X ([X]_i - [X]_e), \\[2ex] \dfrac{d}{dt}q = \alpha_q(1-q) - \beta_q q, \end{cases} \tag{1}$$

where $F$ is Faraday's constant, $z_X$ is the valence of ion $X$, $\lambda_i$ is the osmotic flux rate for compartment $i$, and $q$ denotes the Hodgkin-Huxley gating variables. To model glutamate dynamics in the cleft, we combine existing models of vesicle recycling from Tsodyks and Markram [42] and Walter et al. [43], as illustrated in Fig 2. Non-releasable glutamate pool ($N$) is made readily-releasable through four Ca$^{2+}$-dependent intermediate steps ($R$, $R_1$, $R_2$, $R_3$). After release into the cleft from a fused vesicle, ($F$), glutamate is removed by presynaptic and astroglial EAATs and enters the 'inactive (I)' state. As a final step, glutamate is again stored in a vesicle depot ($D$) and enters the non-releasable pool ($N$) again. The rate equations of these states are given by

$$\frac{d}{dt}N_Y = \sum_X v_X([\text{Ca}^{2+}]_n)N_X, \tag{2}$$

where $Y$ and $X$ span over {$N$, $R$, $R_1$, $R_2$, $R_3$, $I$, $D$, $F$}. The rate constants $v_X$ depend on Ca$^{2+}$ concentrations in the neuronal synaptic compartment. Eqs (1) and (2) describe ion and volume dynamics in the neuronal and astrocyte compartments. To obtain extracellular dynamics, we use three conservation laws, i.e.,

$$\begin{cases} \sum_{X,i} z_X N_X^i = 0, \\[2ex] \sum_i W_i = C_W = \text{constant}, \\[2ex] \sum_i N_X^i = C_X = \text{constant}, \end{cases} \tag{3}$$

for the preservation of charge, volume, and mass, respectively. Our model is essentially described by Eqs (1), (2) and (3). Here, the constants $C_W$ and $C_X$ are the total volume and total

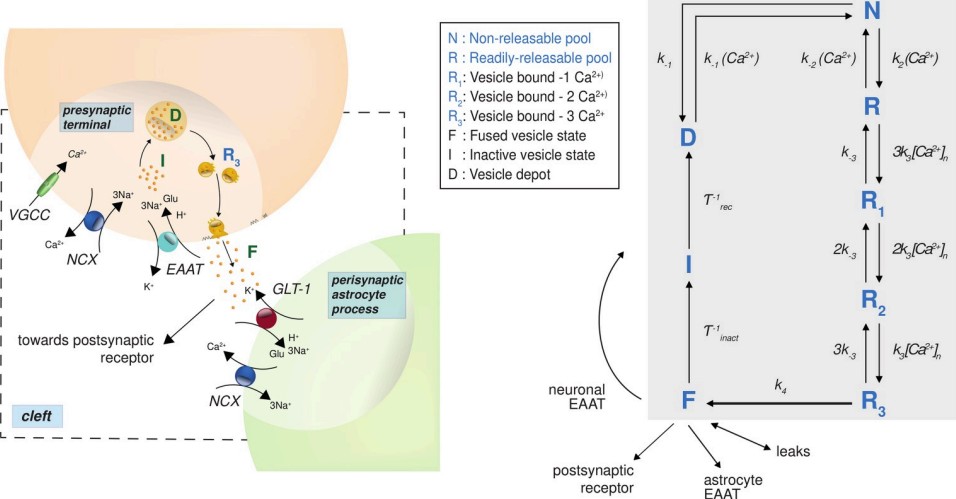

**Fig 2. Glutamate recycling scheme, inspired by combining vesicle-based models from [42] and [43].** (Left) Closer view of the model scheme at the synapse and (Right) the glutamate recycling scheme. Inactive neuronal intracellular glutamate (I) moves to the depot (D) from where it is packed into vesicles which pass through five stages (N, R, $R_{1,2,3}$) before they are released into the synaptic cleft (F). These stages have fast time-constants that depend on intracellular $Ca^{2+}$ concentration. The stages $R_i$ correspond to vesicles that are bound by $i$ $Ca^{2+}$ ions. The time-constants change when there is influx of $Ca^{2+}$ in the presynaptic terminal in response to membrane depolarization. Released glutamate in the cleft can be taken up by astrocytes or back to neurons using excitatory amino acid transporters (EAATs) or leak channels, thereby recycling the released neurotransmitter.

molar amount of ions present in the system, respectively. We present further details in the section **Materials and Methods**.

## Simulations and model calibration

In the following sections, we present and discuss *in silico* experiments simulating ED. We simulate ED by temporary blocking and subsequently restoring of the NKA current in the neuronal and astrocyte compartment. The expression for the NKA current is given by,

$$I_{NKA}^i = \left(\frac{I_{NKA}^{max}(t)}{100}\right)P_{NKA}^{scale}f_{NKA}^i.$$  (4)

The expression $f_{NKA}^i$ describes the inward NKA current. It depends on neuronal $Na^+$ concentration and extracellular $K^+$ concentration. Details can be found in **Materials and Methods**. ED is simulated by manipulating the function $I_{NKA}^{max}(t)$, which is the amount of energy available (in %). The function $I_{NKA}^{max}(t)$ is implemented as

$$I_{NKA}^{max}(t) = P_{min} + (1 - P_{min})I_{block}(t),$$  (5)

where we have the U-shaped function

$$I_{block}(t) = (1 + \exp(\beta(t - t_1)))^{-1} + (1 + \exp(-\beta(t - t_2)))^{-1},$$  (6)

such that

$$t_1 = t_{start} - \frac{1}{\beta}\log(1/P_{min} - 1),$$
$$t_2 = t_{end} + \frac{1}{\beta}\log(1/P_{min} - 1).$$  (7)

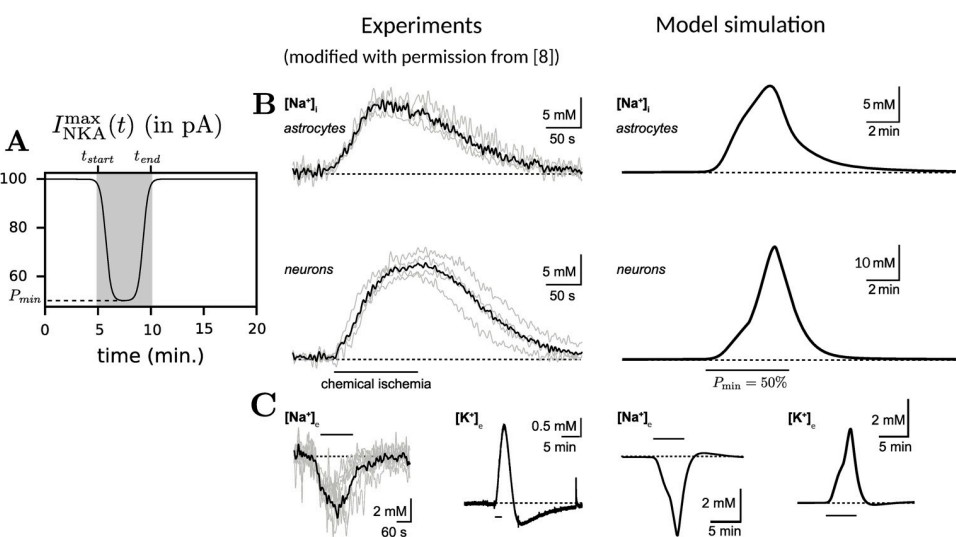

**Fig 3. Model calibrations reproduce experimental data.** (A) Plot of $I_{NKA}^{\max}(t)$ versus time $t$. ED begins at $t = t_{start}$ min. and ends at $t = t_{end}$ min. while being reduced to a minimum of $P_{\min}$. (B,C) Experimental traces from [8] (left) and the corresponding model simulations (right) with (B) showing intracellular sodium for neurons and astrocytes and (C) showing extracellular sodium and potassium. Empirically adjusted parameters: $P_{\min}$ and $\alpha_e^0$ (initial extracellular volume fraction) were chosen by fitting model dynamics qualitatively to $Na^+$ and $K^+$ concentration time-traces. Here, $P_{\min} = 50\%$ and $\alpha_e^0 = 80\%$. The difference in $Na^+$ increase between neurons and astrocytes is attributed to the presence of fast $Na^+$ influx through voltage-gated $Na^+$ channels in neurons, which are lacking in astrocytes. Please note that scaling axis in panels B and C may slightly differ between experiments and simulations for optimal display purposes.

The parameter $\beta$ controls the steepness, $t_{start}$ denotes the onset time, and $t_{end}$ denotes the offset time. The parameter $P_{\min}$ is the minimum available energy when ED is induced. Fig 3A shows how this is implemented.

All the parameters in this model have been sourced from previously published work. However, we set the parameters $\alpha_e$, $P_{\min}$, $P_{NKA}^n (= P_{NKA}^a)$ to fit the experimental traces in [8] using an empirical parameter search. In these experiments, metabolism is inhibited by exposing cells to 'chemical ischemia' for 2 minutes. Recordings are made before, during and after the event. Of note, the transient drop in ATP levels is not abrupt and lasts longer than 2 minutes. To account for this, we simulate ED for 5 minutes and compare to compartmental $Na^+$ and extracellular $K^+$ concentrations, see Fig 3. As details about the change of NKA activity resulting from chemical ischemia used in the experiments are not completely known, we observe some differences in neuronal and astrocytic $Na^+$ gradients during the onset of ED. In neurons, the presence of voltage-gated $Na^+$ channels that are activated upon membrane depolarisation causes a fast and strong $Na^+$ influx. In contrast, the model for astrocytes does not have any fast, channel-mediated $Na^+$ influx pathway, resulting in lower $Na^+$ rise. Thus, we characterize the chemical ischemia experiments from [8] with the parameter values $\alpha_e = 80\%$, $P_{\min} = 50\%$ and $P_{NKA}^n = 86.4$ pA, and use them for further simulations presented in this work.

We summarize the various simulations performed ahead in Table 2. Apart from finite-time ED, we also perform simulations for different values of $P_{scale}$ and $\alpha_e$. This can have consequences on other parameters. $P_{scale}$ scales the baseline NKA strength $P_{NKA}^i$, making it stronger ($P_{scale} > 1$) or weaker ($P_{scale} < 1$). Upon changing $P_{scale}$ from its default value of 1, the equilibrium corresponding to the initial (baseline) conditions disappears. To fix this equilibrium, we recompute all leak permeabilities $P_L^{X,i}$ whenever $P_{scale}$ is changed.

**Table 2. Parameter values for all simulations performed.** Units are presented in the same manner as they are implemented in the Python code.

| Simulation | Parameter values | Description |
|---|---|---|
| Calibration | $P_{\min} = 50\%$ | Parameters used to calibrate to the two-minute *in vitro* chemical ischemia experiments performed in [8]. |
| | $\alpha_e = 80\%$ | |
| | $P_{scale} = 1$ | |
| | $t_{start} = 5$ min. | |
| | $t_{end} = 10$ min. | |
| Excitation | $P_{\min} = 100\%$ | Exciting the neuron in the presence and absence of the astrocyte. |
| | $\alpha_e = 80\%$ | |
| | $P_{scale} = 1$ | |
| | $\max(I_{excite}) = 20$ pA (4 times, 10s) | |
| ED | $P_{\min} = 50\%$ | Transient ED for a) long and short duration, b) small and large surrounding extracellular volume and c) normal and stronger NKA pump strength. |
| | $\alpha_e = 20\%$ (Small ECS) | |
| | $\alpha_e = 80\%$ (Large ECS) | |
| | $P_{scale} = 1$ (normal) | |
| | $P_{scale} = 2$ (strong) | |
| | $t_{start} = 5$ min. | |
| | $t_{end} = 10$ min. (short) | |
| | $t_{end} = 20$ min. (long) | |

The parameter $\alpha_e$ is the initial extracellular volume ratio and is given by

$$\alpha_e = \frac{W_e^0}{W_{\text{tot}}}. \tag{8}$$

We fix the initial volumes $W_n^0$ and $W_a^0$, and choose the extracellular volume to be

$$W_e^0 = \alpha_e(W_n^0 + W_a^0)(1 - \alpha_e)^{-1},$$
$$W_{\text{tot}} = W_n^0 + W_a^0 + W_e^0. \tag{9}$$

## Results

### Astrocytes and ion homeostasis

We first assess the contribution of the astrocyte to the extra- and intracellular ion homeostasis under physiological conditions. For this, we stimulate our neuron with a 10-second long pulse of square wave input with magnitude 25 pA, both with and without a functional astrocyte (Fig 4). Simulations are performed with a realistic initial extracellular volume fraction for mature rodent cortex [36] by setting $\alpha_e = 20\%$ and full ATP availability ($P_{\min} = 100\%$). When the astrocyte is functional, the current injection (black trace) induces a burst of neuronal action potentials (in total 475 during 10 s) and transient depolarizations of the astrocyte. The burst firing of neurons is accompanied by a transient decrease in extracellular Na$^+$, while neuronal Na$^+$ increases, consistent with experimental data [44]. On the other hand, in our simulation, astrocytic Na$^+$ slightly decreases in response to neuronal burst firing. After stimulation subsides, both membrane potentials and ion concentrations return to baseline, see Fig 4A. Profiles of Ca$^{2+}$ and other ion fluxes are presented in S1 Fig.

When the astrocyte is non-functional (simulated by blocking all astrocyte ion transport), the neuron irreversibly depolarizes during the burst (Fig 4B). This new state is accompanied

**Fig 4. Shown are the time courses of the membrane potential, sodium and potassium concentrations, cell volume and glutamate in response to a current pulse (25 pA, 10 s, black trace) with (A) and without (B) a functional astrocyte.** (A) In response to the pulse, there is a burst of action potentials and return to baseline of all quantities. (B) Without a functional astrocyte, the neuron depolarizes after the burst, and remains in this state, even if the astrocyte function is restored. Here, we plot neuronal (blue), astrocyte (orange) and extracellular (green) traces against time for several quantities. The initial extracellular volume ratio is $\alpha_e = 20\%$. The shaded red area corresponds to periods during which ion transport across the astrocytic plasma membrane is blocked.

by a significant accumulation of $Na^+$ in the neuronal soma following $Na^+$ entry via voltage-gated $Na^+$ channels. As astrocytic $K^+$ uptake via Kir4.1, NKA and NKCC1, is blocked, $K^+$ accumulates in the extracellular space, resulting in a sustained depolarization block of the neuron. As the continued $Na^+$ influx is higher than NKA-mediated $Na^+$ efflux, the neuron accumulates even more $Na^+$ and does not recover even after excitation ends. This irreversibly depolarized state is accompanied by an increased volume of the intracellular compartments resulting from an increase in intracellular $Na^+$ and $Cl^-$.

Upon restoring astrocytic function, the neuron and astrocyte do not return to their initial resting membrane voltage (Fig 4B). Membrane potentials remain at approximately -30 mV, and input currents can no longer trigger action potentials. From the two simulations performed Fig 4A and 4B, we see that two stable resting states are possible, i.e., the system displays *bistability*. The first state corresponds to membrane potentials of approximately -65 mV (neuron) and -80 mV (astrocyte), respectively, with functional synaptic transmission, i.e. the physiological state. The second corresponds to a pathological state with non-functional neurons and astrocytes, and impaired synaptic transmission.

## Dynamics after varying periods of ED

We know from experimental observations that energy depletion results in accumulation of $Na^+$ in neurons and astrocytes while extracellular $K^+$ increases see, e.g. [8, 12]. At the (thermodynamic) equilibrium, the Nernst potentials of $Na^+$ and $K^+$ reach equal values. During this evolution towards equilibrium, neurons can generate oscillatory depolarizations, known as anoxic oscillations [4, 7]. These drive $Ca^{2+}$ into the presynaptic terminal by activating of voltage-gated $Ca^{2+}$ channels (VGCCs) which triggers the release of glutamate into the cleft. The efflux of $K^+$ does not fully compensate the increase in intracellular $Na^+$. Thus, intracellular $Cl^-$ increases to preserve electroneutrality [45]. The resulting osmotic imbalance leads to water influx, i.e., cell swelling [12].

We first simulate such dynamics by setting the available energy $P_{min}$ to 50% of baseline activity at an initial extracellular volume ratio $\alpha_e$ = 80% for 5 minutes, shown in Fig 5A. Similar to experimental observations, reduced NKA activity leads to an increase in intracellular $Na^+$ and extracellular $K^+$. Reactivated NKA restores the $Na^+$ and $K^+$ gradients, so membrane potentials recover to baseline in agreement with experiments [8, 32]. Changes in cell volume are below 5%, i.e., they are negligible for this simulation.

However, when we extend the duration of ED from 5 to 15 minutes, $Na^+$ accumulation in the neuron is much larger than in the previous scenario. A depolarization block results, where the membrane potential approaches a stable pathological state of about -35 mV. As shown in Fig 5B, the increase in intracellular $Na^+$ and $Cl^-$ (see S2 Fig) causes an osmotic imbalance resulting in cell swelling. $Na^+$ accumulation in the astrocyte reduces EAAT activity, preventing successful clearance of excess cleft glutamate. Upon restoring energy, astrocytic $Na^+$ recovers and enables EAAT to drive glutamate into the astrocyte. At the same time, in contrast, neuronal EAAT activity weakens, which prevents sufficient reuptake of glutamate into the presynaptic terminal. As a result, during ATP depletion, transport of glutamate into presynaptic vesicles is initially diminished, see neuronal EAAT current in S2 Fig. Further, ion gradients do not return to physiological values, and permanent cell swelling is observed (approximately 20% increase in neuronal, and 10% in astrocytic volume). This occurs once again due to bistability. After a sufficiently long, but transient, period of ED resulting in both neuron and astrocytic reduction of NKA activity, the physiological state is not restored, as reflected by persistent membrane depolarization and increased cell volume. This contrasts with the scenario

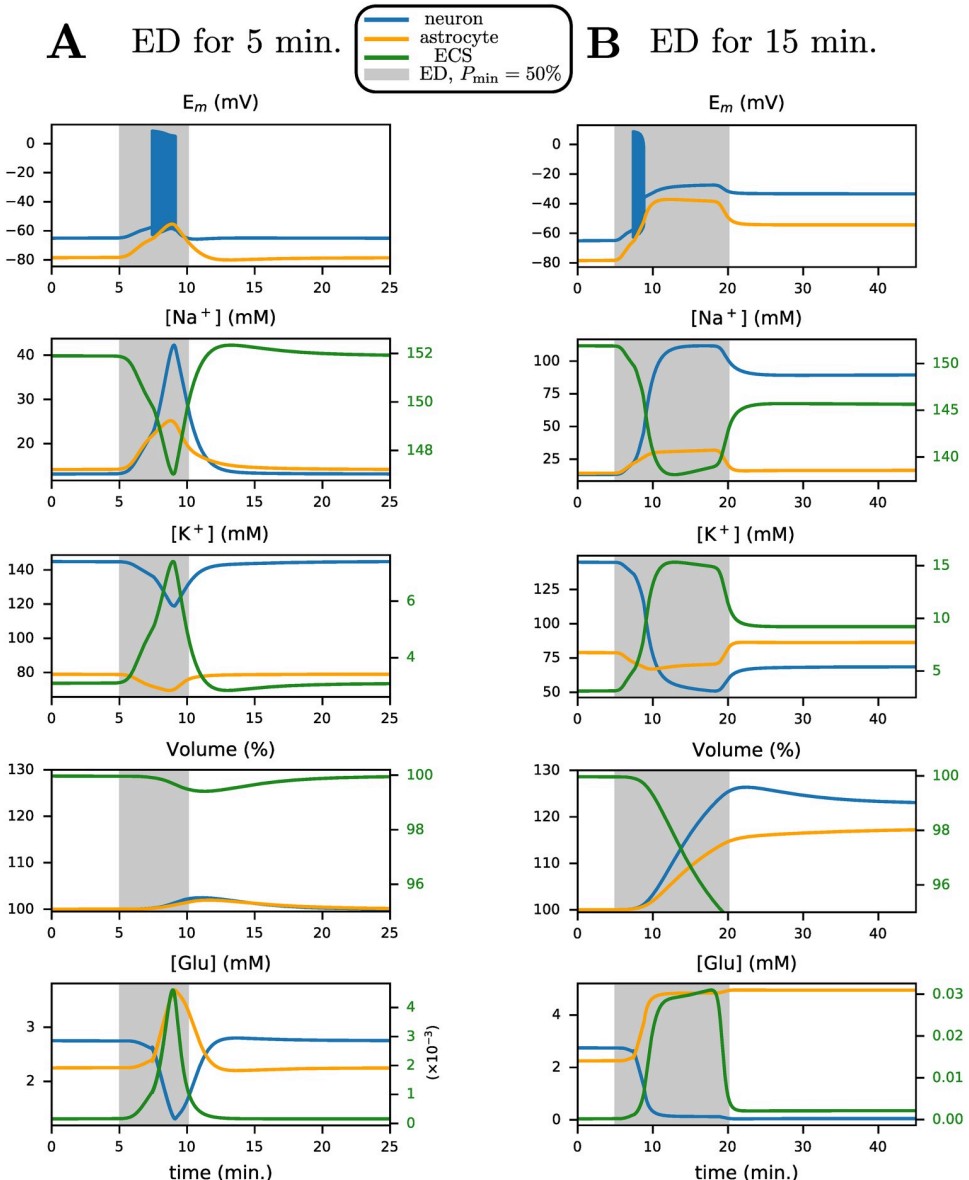

**Fig 5. ED for 5 minutes (A) and 15 minutes (B) demonstrates the existence of two stable states: 1) before ED (baseline resting state) and 2) after prolonged ED of 15 minutes (stable depolarized state).** Here, we plot neuronal (blue), astrocyte (orange) and extracellular (green) traces against time for several quantities. The initial extracellular volume ratio $\alpha_e$ = 80% and minimal energy available $P_{min}$ = 50%. Shaded grey areas correspond to the period where Na$^+$/K$^+$-ATPase (NKA) activity is gradually reduced to $P_{min}$ and restored to baseline, identical to Fig 3A.

presented in Fig 5A, where the physiological state is restored for a shorter duration of ED. Time-traces for concentrations of Cl$^-$ and Ca$^{2+}$ are presented in S2 Fig.

## Determinants for recovery

Various determinants other than the depth and duration of ED have been proposed to be critical for recovery of synaptic transmission failure after ED, including the size of ECS, pump strength, and the role of the gating variables. Using our model, we explore their importance in the potential of neuronal recovery after (transient) ED.

**Size of the extracellular volume.** Recovery from a depolarization block after energy restoration was addressed experimentally in [32]. Neurons from two different brain regions were subjected to 15 minutes of zero energy conditions. Magnocellular neuroendocrine cells recovered, while pyramidal cells remained depolarized post energy restoration (see also Fig 8). As argued in [46], this may result from different ECS volume fractions. We now explore this hypothesis in our model by changing the baseline value of the parameter $\alpha_e$ (the extracellular volume fraction in %). In Fig 6A, we simulate ED for 5 minutes for $\alpha_e = 80\%$ (large ECS) and

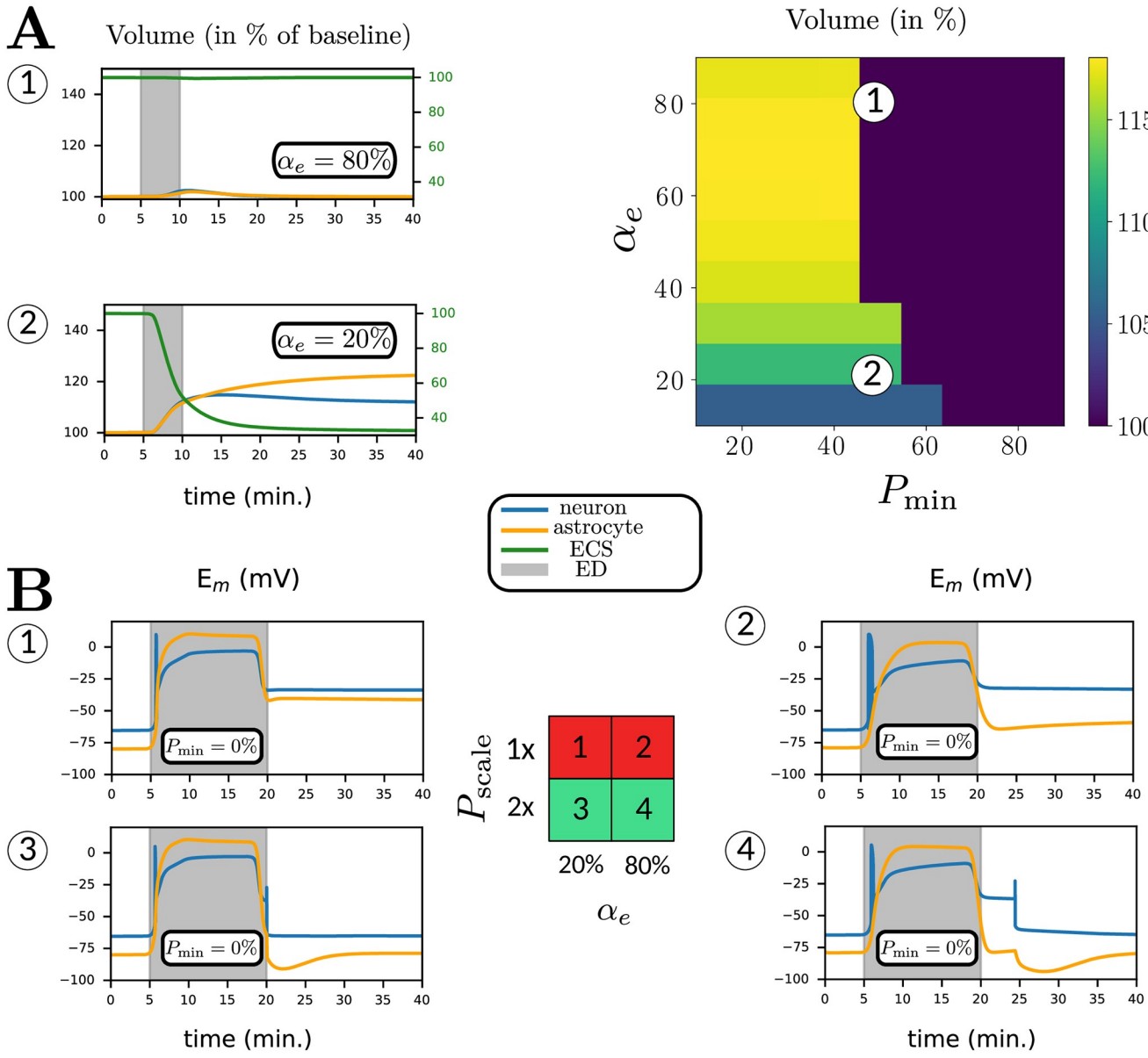

**Fig 6. Differential sensitivity of ED to initial extracellular volume ratio $\alpha_e$ (A) and to baseline Na$^+$/K$^+$-ATPase strength factor $P_{\text{scale}}$ (B).** (A) We deprive the neuron and astrocyte of energy for 5 minutes before restoring it to baseline and report the relative volume 30 minutes after restoration. We show two examples, (A.1) for large ECS ($\alpha_e = 80\%$) and (A.2) for realistic ECS ($\alpha_e = 20\%$). Here, we plot neuronal (blue), astrocyte (orange) and extracellular (green) traces against time for compartmental volume change. (B) We deprive the neuron and astrocyte of energy for 15 minutes before restoring it to baseline, for two different values of $P_{\text{scale}}$ and $\alpha_e$. We show neuronal and astrocyte membrane potentials against time. The grey area in (B.1–4) illustrates the period of ED. The table in the middle indicates whether the system recovers (green) post energy restoration or not (red).

$\alpha_e$ = 20% (small, realistic ECS) and plot the relative neuronal volume change after 30 minutes of energy restoration. The parameter $P_{min}$ is set to 50%. For $\alpha_e$ = 80%, we observe no volume increase, which corresponds to a successful recovery after energy restoration, and vice-versa for $\alpha_e$ = 80%. Further, we simulate ED during 5 minutes for different values of $\alpha_e$ and $P_{min}$ and plot the relative neuronal volume change after 30 minutes of energy restoration. We find that compartmental volumes (and thus, membrane potentials and ion concentrations) recover for larger values of $\alpha_e$. This result implies that neurons and astrocytes surrounded by smaller extracellular spaces exhibit a relatively higher vulnerability to transient ED.

We provide further insight into the selective sensitivity to ECS size using bifurcation diagrams shown in Fig 7A. We show diagrams for both $\alpha_e$ = 20% and $\alpha_e$ = 80% as a function of the available energy $P_{min}$. We plot initial neuronal volume (in %) against $P_{min}$, after setting $I_{NKA}^{max}(t) = P_{min}$ as constant. We obtain two branches of resting states. Solid (red/blue) lines correspond to stable resting states, while the dashed blue branches correspond to unstable resting states.

The bifurcation diagram can be interpreted by following the stable resting states. We start at the blue line at $P_{min}$ = 100%, which corresponds to the baseline physiological state. As we lower NKA activity in the neuron and astrocyte, we move to the left, along the solid blue curve. For values lower than $P_{min}$ = 63.93%, the blue curve does not exist. Decreasing $P_{min}$, the physiological branch disappears via a limit point bifurcation (star). For $\alpha_e$ = 80%, the limit point occurs at $P_{min}$ = 57.4%. At this bifurcation point, unstable (dashed blue line) and stable (solid blue line) resting states merge. For lower values of $P_{min}$, to the left of the limit point, the pathological state is the only stable resting state. The loss of bistability here explains the transition to the pathological state during ED.

This is further illustrated with two simulations where energy is deprived for 5 (turquoise trace) and 15 minutes (magenta trace) for $P_{min}$ = 50%. These simulations are presented simultaneously in the bifurcation diagram (Fig 7A.1a and 7A.2a) and the time traces (Fig 7A.1b and 7A.2b). When energy is deprived for 5 minutes, the system transits to the pathological state for $\alpha_e$ = 20% and does not recover to the physiological state. For $\alpha_e$ = 80%, the same simulation results in recovery. However, upon restoring energy later (15 minutes, orange trace), the synapse transits to the pathological resting state for both values of $\alpha_e$. During energy restoration, the system's state slides along the red curve. When we return to $P_{min}$ = 100%, the pathological state is still stable, implying that recovery is impossible without further intervention. As the pump is already fully activated, this resting state would become unstable only if its maximal strength $P_{min}$ is increased above 145% of its initial value, at which a Hopf bifurcation occurs (inverted triangle), which is biophysically unrealistic. The pathological and physiological resting state curves are much farther apart for $\alpha_e$ = 80% than for $\alpha_e$ = 20%. Moreover, for $\alpha_e$ = 80%, the limit point bifurcation (star) is further to the left (i.e. more severe ED). This shows that the transition to the pathological state occurs at larger values of $P_{min}$ for synapses with a small ECS (small $\alpha_e$).

**Pump strength.** Next, we explored the consequences of different pump strengths on the recovery of membrane potentials and ion dynamics after transient ED. We mimic this behaviour by changing the baseline maximum NKA current. We thus change the parameter $P_{NKA}^{scale}$. At baseline, $P_{NKA}^{scale} = 1$. When $P_{NKA}^{scale} > 1$, NKA becomes stronger by a factor $P_{NKA}^{scale}$. We keep the baseline conditions (ion concentrations, volumes and membrane potentials) fixed by changing the leak permeabilities of $K^+$ and $Na^+$ in neurons and astrocytes.

We first simulate ED for 15 minutes in large ECS ($\alpha_e$ = 80%) and realistic ECS ($\alpha_e$ = 20%), shown in Fig 6B.1 and 6B.2. Here, we set $P_{NKA}^{scale}$ to 1 and $P_{min}$ to 0%. In both cases, neuronal and astrocyte membranes depolarize and do not recover after energy is restored. Note that it takes

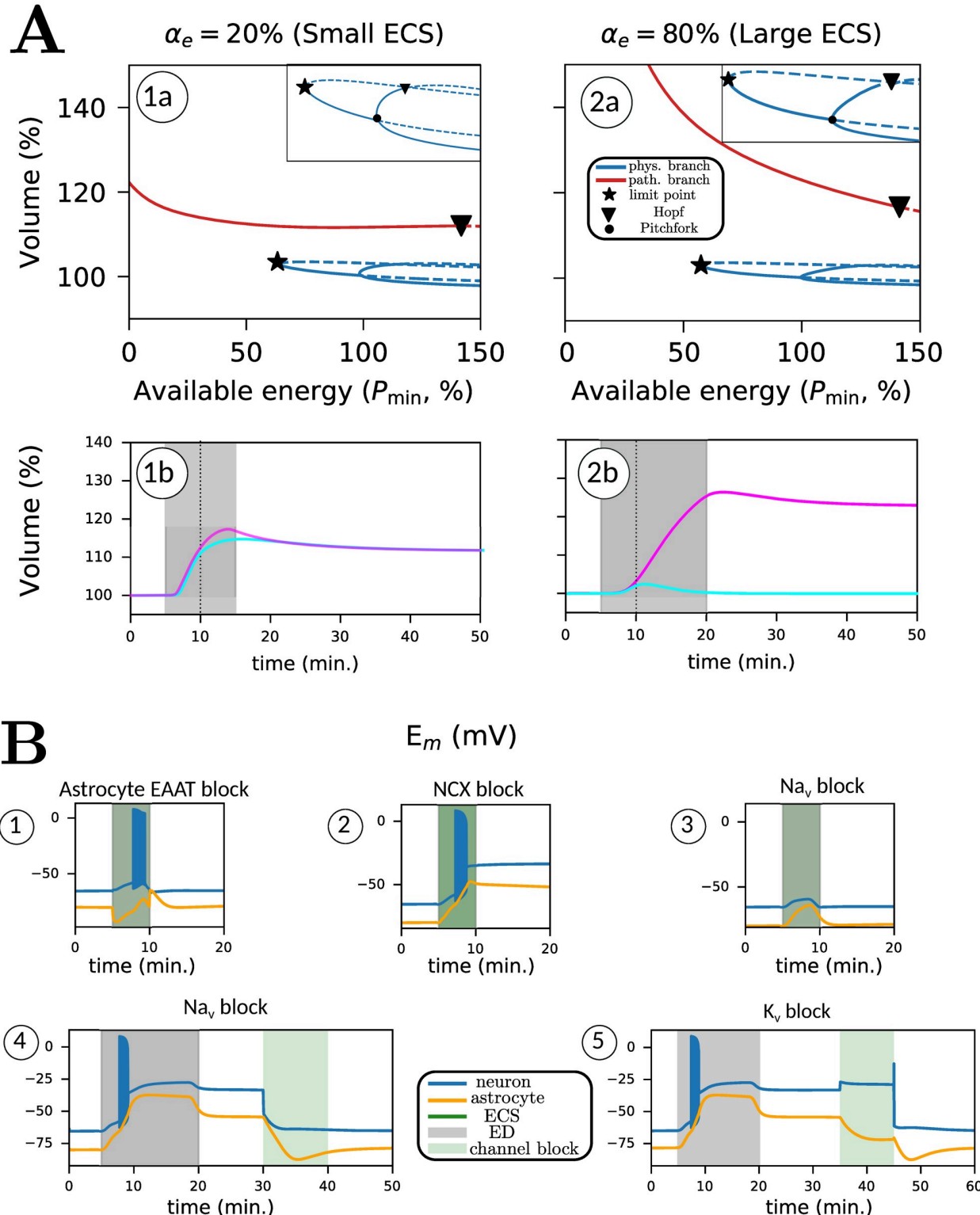

**Fig 7. Tipping in a *bistable* regime (A) and the change in tipping behavior by introducing pharmacological blockers (B).** In (A.1a and A.2a), we plot bifurcation diagrams with respect to $P_{min}$ for $\alpha_e = 20\%$ (realistic ECS) and $\alpha_e = 80\%$ (large ECS). Red curves are pathological branches, and blue curves are physiological branches. Dashed lines represent unstable parts. The only two relevant local bifurcations are limit point (star) and Hopf (inverted triangle). The inset shows two additional bifurcations, a pitchfork (dot) and a Hopf. We show two simulations (A.1b and A.2b), short ED (5 minutes, cyan curve) and long ED (15 minutes, pink curve), both for $P_{min} = 50\%$. In (B), we block different pathways during ED (green area, B.1,

B.2 and B.3) and after restoration (green area, B.4). For (B.1–3), energy is deprived for 5 minutes for parameters $\alpha_e$ = 80% and $P_{\min}$ = 50%. In (B.4), energy is deprived for 15 minutes for parameters $\alpha_e$ = 80% and $P_{\min}$ = 50%.

longer to reach the peak depolarization for neurons in large ECS, in agreement with observations in the bifurcation diagram (Fig 7A). In Fig 6B.3 and 6B.4, we show results of the simulations with $P_{\mathrm{NKA}}^{\mathrm{scale}} = 2$. This corresponds to NKA being twice as strong. In both cases, the membrane potentials transit to a stable depolarization block during the period of ED. In both cases, the synapse now returns to the physiological resting state after energy restoration. In the case of $\alpha_e$ = 80%, the neuronal membrane potential closely mimics experimental findings for magnocellular neuroendocrine cells [32] (see Fig 8B). We also observe that the neuron undergoes transient spiking immediately before recovery. This spiking results from a Hopf bifurcation, shown in the corresponding bifurcation diagram (Fig 7A). When $P_{\mathrm{NKA}}^{\mathrm{scale}} > 1$, the Hopf bifurcation shifts to the left (not shown). In this case, the corresponding Hopf bifurcation has shifted left of $P_{\min}$ = 100%. The Hopf bifurcation is supercritical, which implies that a stable periodic orbit is born. This periodic orbit disappears before $P_{\min}$ = 100%. Thus the neuron generates action potentials while it transits through the small parameter regime where the periodic orbit exists (Fig 6B.4). When $I_{\mathrm{NKA}}^{\max}$ reaches 100%, the periodic orbit disappears, and the membrane potential returns to baseline.

**Pharmacological blockers.** We simulate various scenarios observed experimentally where neurons and astrocytes in acute tissue slices of mouse brain were subjected to ED in the presence of an inhibitor of EAATs (TFB-TBOA) or NCX [8]. We first set $\alpha_e$ = 80% and $P_{\min}$ = 50%, identical to Fig 5, and present four scenarios, illustrated in Fig 7B. First, we block EAAT transport (B.1) during the duration of ED, which blocks EAAT-mediated $Na^+$ influx into astrocytes. As a result, neurons and astrocytes depolarize before returning to baseline. Second, blocking NCX transport (B.2) has a stronger effect on membrane potentials, and both cells do

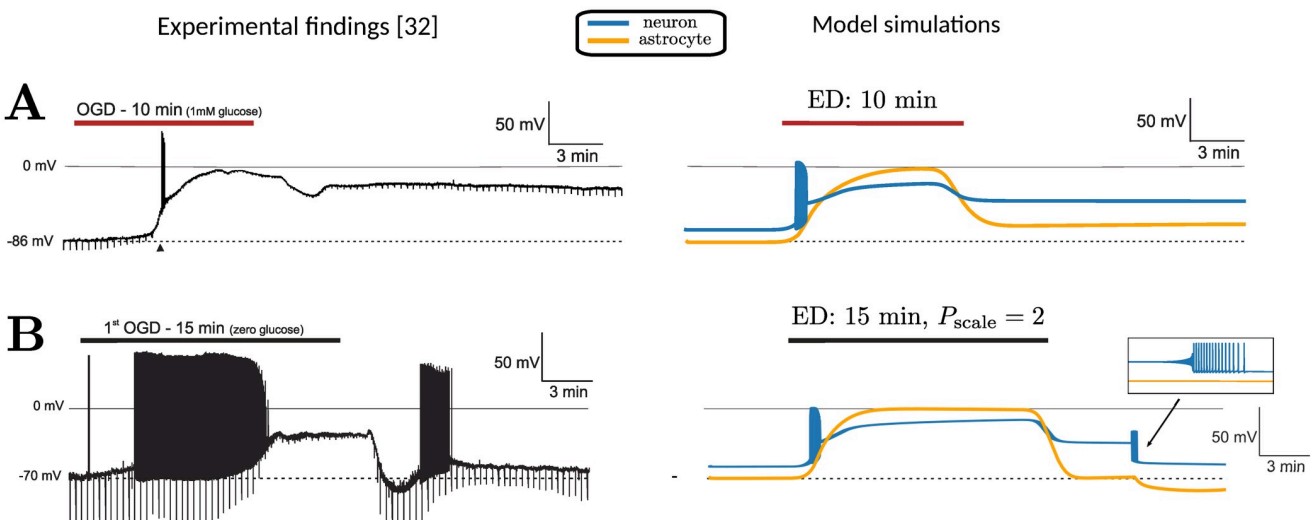

**Fig 8. Experimental findings from Brisson and Andrew [32] (reproduced with permission) and our model simulations.** (A) (Left) Membrane depolarization of a pyramidal neuron during 10 minutes of oxygen glucose deprivation (OGD), that persists after restoring energy. (Right) Model simulations on the right show the neuronal (blue) and astrocyte membrane potentials. ED (OGD) is introduced for 15 minutes (red line). Here, $\alpha_e$ = 80% and $P_{\min}$ = 0%. The dynamics are faithfully reproduced, including anoxic oscillations at the initial phase of depolarization. (B) (Left) Membrane potential of a magnocellular neuroendocrine cell, showing a similar depolarization during 15 minutes oxygen glucose deprivation and full recovery after this period. Both at the start of the depolarization and during recovery action potentials are generated. (Right) Model simulations. ED (OGD) is introduced for 15 minutes (black line). Here, $\alpha_e$ = 80%, $P_{\min}$ = 0% and $P_{\mathrm{scale}}$ = 2. With these parameter settings, the membrane potential recovers to baseline conditions after the period with ED. Note that during recovery, the experimentally observed oscillations are also faithfully simulated.

not recover after restoring energy. During ED, NCX reverses, and thus mediates $Na^+$ efflux in low energy conditions, a phenomenon also suggested in experimental observations [8]. Blocking NCX transport promotes the cellular depolarization of the neuron and the astrocyte, driving them to the irreversible pathological state.

Third, motivated by experimental evidence [12] and modelling work [11], we explore the potential for recovery after blocking neuronal $Na^+$ influx pathways. We first simulate a short duration of ED ($P_{min}$ = 50%) simultaneously blocking the voltage-gated $Na^+$ channel, upon which the neuron barely depolarizes (Fig 7B.3). Next, we consider the system after it has transited to a stable pathological state after longer ED, shown in (Fig 7B.4). Subsequently, we block the voltage-gated $Na^+$ channel of the neuron for 10 minutes. We observe that the neuron repolarizes and transits to the physiological resting state, with the restoration of all other ion concentrations and cell volume.

Motivated by this, we block neuronal $K^+$ efflux pathways to explore the potential for recovery. As before, we consider the system after it has transited to a stable pathological state, after longer ED. Blocking the voltage-gated $K^+$ channel for 10 minutes (Fig 7B.4), which is the major $K^+$ efflux pathway in the neuron, results in neuron and astrocyte repolarisation, and the entire system transits to the stable physiological resting state, as before ED.

In Fig 9, we demonstrate synaptic recovery after blocking the voltage-gated $Na^+$ channel and voltage-gated $K^+$ channel. We set $\alpha_e$ = 80% and subject the neuron to ED for 15 minutes ($P_{min}$ = 50%). Then, as the neuron has transited to the pathological state, between $t$ = 30 and $t$ = 40 minutes, we stimulate the cell with 25 pA current for 10 seconds, identical to Fig 4. Glutamate transients in the cleft during this period are shown in Fig 9A.1 ($Na^+$ channel) and Fig 9B.1. In both cases, the relative change in glutamate is about 10%, which is less than the change in glutamate transients in the physiological state (about 30%), shown in Fig 4A. Next, we block voltage-gated $Na^+$ and $K^+$ channels for 10 minutes (green region), bringing the system back to the physiological state. We then test synaptic recovery by applying the same stimulation to the neuron, 70 minutes after the block. The stimulation now induces a burst in the neuron and glutamate transients in the cleft, identical to Fig 4A, thereby demonstrating restoration of physiological synaptic function.

Finally, we mention that, apart from neuronal $Na^+$ and $K^+$ voltage-gated channels, blocking any other ion channel did not assist in recovery from the pathological state, see S3 Fig.

## Discussion

We present a detailed biophysical model of energy-dependent ion fluxes in different compartments and of changes in cellular membrane potentials of the tripartite synapse to further our understanding of their dynamics in low energy conditions. We calibrate the model to $Na^+$ and $K^+$ concentration time-traces obtained from in-situ chemical ischemia experiments [8]. We demonstrate that astrocyte function is instrumental in maintaining physiological ion gradients for action potential generation and proper synaptic transmission. Crucially, the model indicates that surrounding extracellular volume size and baseline NKA pumping capability controls the ischemic vulnerability of the neuron-astrocyte interaction. Further, bifurcation analysis shows how the bistability depends on extracellular volume. Finally, we show that intervention through blocking voltage-gated $Na^+$ channels can revive the system from a pathological state.

### Loss of synaptic function depends on the depth and duration of ischemia

In resting conditions, our model shows astrocyte and neuron membrane potentials close to the $K^+$ Nernst potential, similar to experimental observations. Loss of NKA-mediated $Na^+$ and

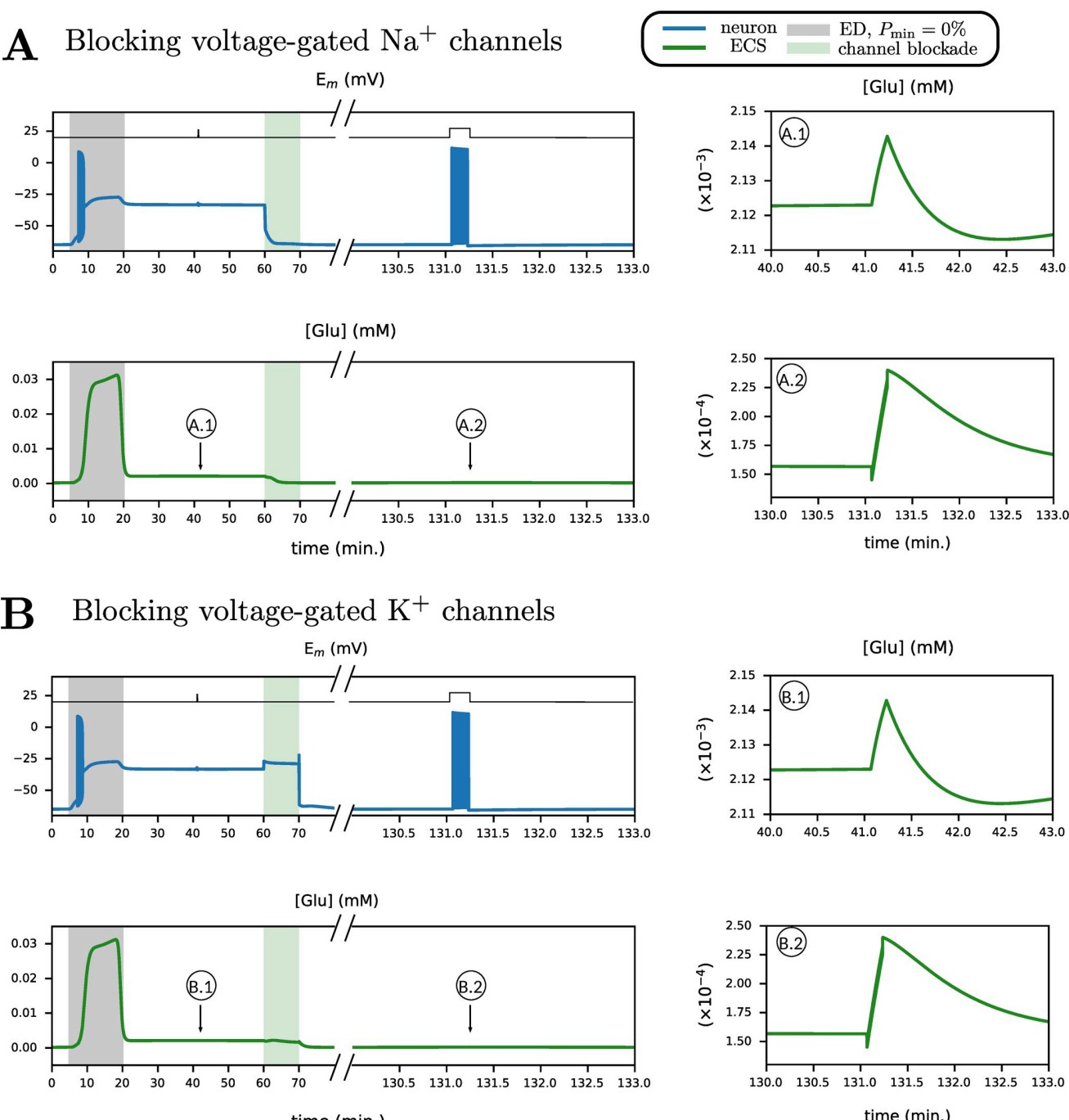

**Fig 9. Neuronal stimulation upon recovery from ED shows different glutamate transients as compared to neuronal stimulation in a pathological state.** Recovery from ED is achieved by blocking neuronal voltage-gated Na$^+$ channels (A) and blocking neuronal voltge-gated K$^+$ channels (B). Glutamate in the cleft (green trace) and neuronal membrane potential (blue trace) are shown, in response to neuronal excitation in physiological and pathological conditions. First, ED is simulated between $t = 5$ and $t = 20$ minutes ($P_{min} = 50\%$, $\alpha_e = 80\%$). Then, neurons are subjected to 25 pA square wave input for 10 seconds, as indicated by the black trace. The system is then brought back to the physiological state by blocking voltage-gated Na$^+$ channels (shaded green area). After a little more than an hour, the neurons are subjected again, to 25 pA square wave input for 10 seconds.

$K^+$ transport results in intracellular $Na^+$ and extracellular $K^+$ accumulation and membrane depolarization, in accordance with experimental observations [8, 32] and previous simulations [11]. As expected [47], astrocytic dysfunction in our simulations also results in disruption of $K^+$ homeostasis and cessation of synaptic transmission, even in conditions with initially preserved neuronal function. However, upon neuronal stimulation, astrocytic $Na^+$ decreases. Earlier experimental work performed in brain tissue slices [44] has shown that most astrocytes undergo an increase in intracellular $Na^+$ during neuronal bursting activity. However, in about a third of cells, the increase was rather brief and was followed by an undershoot in $[Na^+]_i$. This undershoot could be mimicked by increasing extracellular $K^+$, indicating that it was due to activation of astrocyte NKA. Moreover, the combined addition of glutamate and high $K^+$ caused a biphasic increase-decrease in some cells, indicating that both processes—activation of NKA resulting in an export of $Na^+$ and activation of glutamate uptake resulting in an import of $Na^+$—counteract each other. Notably in our model, NKA-induced export of $Na^+$ is higher than EAAT-mediated $Na^+$ uptake in astrocytes.

If the partial and transient ED is sufficiently long, no recovery occurs. During low energy conditions, glutamate in the cleft can rise to several hundred µM or even a few mM [48] and excitotoxic cell death will eventually follow [16]. The persistent increase in glutamate during the duration of ED is faithfully reproduced in our simulations (Fig 5B).

Our model simulations during low energy conditions also agree with the experimental findings from [32], shown in Fig 8: during oxygen-glucose deprivation (OGD) neurons depolarize and cease firing action potentials after a short period of anoxic oscillations. We further show that the neuron can show bistable behaviour: even after cessation of OGD (simulated by restoring pump activity), the neuron can remain in a depolarized, pathological, state (Fig 8A).

## Ischemic vulnerability depends on the extracellular volume fraction and baseline pump strength

Our simulations in Fig 6 show that a smaller ECS makes the neurons and astrocytes more likely to depolarize as the extracellular ion gradients will change faster; further, a smaller ECS makes it more likely that neurons and astrocytes remain in the pathological state, which is further illustrated by the bifurcation diagrams in Fig 7. We conclude that synapses surrounded by smaller ECS are more vulnerable to ED. For a larger ECS, the basins of attraction are farther apart in state space, making it less likely for the trajectory to escape from the physiological equilibrium to the pathological one. These simulations predict that synapses with smaller extracellular spaces are less likely to recover from ED. As the ECS size significantly declines with ageing [33, 36], this may aggravate ischemic damage to the aged brain [49].

We also simulated the recovery observed in magnocellular neurons by changing the baseline pump strength with the parameter $P_{NKA}^{scale}$ in Fig 8B. The neuron makes a few transient oscillations before returning to baseline, which was indeed observed in [32]. In this case, the Hopf bifurcation (inverted triangle) in Fig 7A shifts further to the left. Increasing baseline pump strength makes the physiological state the only stable resting state at baseline by moving the position of the Hopf bifurcation. The importance of baseline pump strength may also explain the experimental observation of Brisson and Andrew, showing that transient oxygen-glucose deprivation has differential effects on the potential for recovery of neurons in the hypothalamus and thalamus [32]. In similar experiments [50, 51], the difference in NKA activity due to varying $\alpha$-isoform expression was highlighted as a possible governing factor for recovery.

We remark that our findings are at variance with an earlier modelling study, showing that smaller ECS size improves the recovery of neurons from transient ATP depletion [46]. In this model [46], that comprised $Na^+$, $K^+$ and $Cl^-$ currents of a presynaptic neuron and the $Na^+/K^+$-

ATPase, the role of glia was limited to $K^+$ buffering and passive extracellular $K^+$ diffusion. This modelling choice may underlie the divergent findings in our simulations, where an extended role for astrocytes together with ECS volume restrictions allows persistent $K^+$ accumulation in the ECS.

## Neurons can be rescued from the pathological equilibrium state

In agreement with experimental observations [32], our simulations also show that a pathological state may persist after energy restoration. Phenomenologically, this persistence results from a persistent $Na^+$ current (the window current) at a membrane voltage near -30 mV [11, 52], that is too large to be counteracted by the $Na^+/K^+$ pump. The system can be rescued from this state, however, by a temporary blockage of voltage-gated $Na^+$ and $K^+$ currents, see Figs 7B and 9. We show that blocking $Na^+$ influx via voltage-gated $Na^+$ channels and $K^+$ efflux via voltage-gated $K^+$ channels serves to recover neuronal and astrocytic membrane potentials and $Na^+/K^+$ homeostasis. Moreover, in this recovered state, healthy synaptic function is resumed as demonstrated by exciting the neuron in Fig 9, where glutamate transients also return to the physiological range, identical to Fig 4A.

Our prediction regarding voltage-gated $Na^+$ channels agrees with previous modelling work, see [11]. While a study in rats after middle cerebral artery occlusion showed significantly improved neurological outcome after treatment with the $Na^+$ channel blocker valproic acid [53], to our knowledge, no experimental data have been reported that explicitly support this model prediction. Our prediction regarding voltage-gated $K^+$ channels is novel with respect to current modelling literature. An experimental study showed that using the $K^+$ channel blocker tetraethylammonium (TEA) was able to attenuate ischemia-triggered apoptosis in neurons [54]. However, to our knowledge, similar to the case with voltage-gated $Na^+$ channels, there is no experimental data that directly supports our prediction regarding $K^+$ channels.

Decrease in NKA activity in neurons and astrocytes creates an imbalance of $Na^+$ and $K^+$ gradients, causing net compartmental $Na^+$ influx and $K^+$ efflux. In that context, it is not too surprising that our model predicts that blocking major $Na^+$ and $K^+$ pathways in the neuron presents a potential pathway for recovery from the pathological state post ED. However this is not the case for astrocytes. Blocking any of the $Na^+$ or $K^+$ pathways post ED in the astrocyte (see supplementary figure S3 Fig) does not perturb the pathological state. This can be attributed to the fact that in our model, there is no astrocytic $Na^+$ influx or $K^+$ efflux process that contributes to large gradients, compared to the neuronal voltage-gated channels.

## Relation to other computational models

Our work extends the single-neuron formalism in [11] to a neuron-astrocyte interaction that describes biophysical processes in synaptic and somatic compartments. There have been several computational studies of neuronal dynamics in the context of energy failure [7, 26, 28, 29], while other studies have explicitly modelled astrocyte dynamics in the context of spreading depression, astrocyte $Ca^{2+}$ signaling and physiological function [22, 40, 55]. To the best of our knowledge, however, no other computational study has explicitly modelled astrocyte dynamics nor did they include $Na^+$, $K^+$, $Cl^-$, $Ca^{2+}$ and glutamate dynamics into a single model in the context of ED. In our work, we combine the dynamics of these five ions to provide a holistic description of ion and volume dysregulation during low energy conditions. Further, the model considers physical laws that arise in limiting cases, such as the Gibbs-Donnan equilibrium, which is reached when pump activity is absent. This provides a platform to extend the current formalism by introducing more ions, cellular compartments and transport mechanisms.

## Modelling limitations

Our model has certain limitations, too. First, we assume spatially uniform compartments and ignore transmission delays, which is not the case in reality. We do not consider the diffusion of ions across the extracellular space, which extends to other synapses as well. Simulating in a finite volume may cause exaggerated extracellular concentrations to appear, which speeds up the pathological effects of ED. However, this will not affect the bifurcation diagram structure, but only how tipping between the two basins of attraction occurs during short term ED. We also do not include astrocyte gap junctions which may change their permeability during low energy conditions and operate in the same timescale as membrane depolarization [56]. However, their role in mitigating extracellular $K^+$ uptake is debated.

## Future directions

The current formulation of the model incorporates ion transients in synaptic and somatic compartments and predicts bistable behaviour in response to ischemia. Adding another neuron with a postsynaptic terminal will allow us to make predictions regarding synaptic transmission in ischemic conditions and to compare presynaptic versus postsynaptic vulnerability to such conditions. The first predictions from our model about glutamate transients can benefit from models such as [57] to include the glutamate-glutamine cycle, which is critical to the dynamics of neurotransmitter replenishment. Moreover, by introducing pH regulation, also the sodium-bicarbonate cotransporter (NBCe1) [58] and the sodium-proton exchanger (NHE) could be incorporated. These mechanisms will increase $Na^+$ transport into astrocytes, and might result in a net increase in astrocytic $Na^+$ with physiological stimulation, improving the current model.

## Conclusion

We introduced a model of the tripartite synapse that describes ion and volume dynamics at synaptic and somatic levels. The model faithfully reproduces biological observations and identifies potential treatment targets to limit permanent synaptic failure in clinical conditions characterized by temporary energy failure. The model predicts that synapses surrounded by smaller extracellular spaces are more vulnerable to ischemia and that differential expression of baseline NKA may explain regional differences in ischemic vulnerability. Further, it predicts that blocking neuronal voltage-gated $Na^+$ and $K^+$ channels rescues the synapse from the pathological state post transient ED. Moreover, the model allows us to include additional processes and combine with more models from current literature to create a mathematical description of critical events concerning synaptic regulation in physiological and pathological conditions at the tripartite synapse.

## Materials and methods

### Ion concentrations and membrane potentials

The model describes the dynamics of molar amounts $N_X^i$ of the ions $Na^+$, $K^+$, $Cl^-$, $Ca^{2+}$ and glutamate, and compartmental volumes $W_i$, for $i = \{n, a\}$. The corresponding membrane potentials follow from the relation,

$$V_i = \frac{\text{F}}{\text{C}_i} \sum_X z_X N_X^i, \tag{10}$$

where $i = \{n, a\}$ and $z_X$ is the valence of the ion/species $X$. The ions $X$ also include impermeable ions $A^-$ and $B^+$ in each of the somatic compartments. This is necessary to maintain a non-zero

resting membrane potential across the semi-permeable neuronal and astrocyte membranes. The molar quantities of these ions are unknown, we estimate them from baseline conditions in the section **Estimating parameters**. Assuming that $Na^+$, $K^+$ and $Cl^-$ concentrations are the same in the somatic and synaptic compartments, we get

$$
\begin{aligned}
[X]_n &= N_X/W_n = [X]_{ps}, \\
[X]_a &= N_X/W_a = [X]_{pap}, \\
[X]_e &= N_X/W_e = [X]_c, \qquad X \in \{Na^+, K^+, Cl^-\}.
\end{aligned}
\tag{11}
$$

The volumes $W_{ps}$, $W_c$, $W_{pap}$ are constant. We further assume that all glutamate and $Ca^{2+}$ in the neurons and astrocytes are located in the synaptic compartments. Thus,

$$
\begin{aligned}
[Y]_n &= N_Y/W_{ps} = [Y]_{ps}, \\
[Y]_a &= N_Y/W_{pap} = [Y]_{pap}, \\
[Y]_e &= N_Y/W_c = [Y]_c, \qquad Y \in \{Ca^{2+}, Glu\}.
\end{aligned}
\tag{12}
$$

In the following sections, we elaborate on how the dynamics of ion amounts $N_X^i$ an volumes $W_i$ are described in the model. The values of a few common parameters, such as fixed volumes and physical constants, are presented in Table 3.

## Neuronal dynamics

The following currents/fluxes are used to describe neuronal somatic dynamics:

1. voltage-gated $Na^+$, $K^+$, $Cl^-$ and $Ca^{2+}$ channels,

2. $Na^+/K^+$-ATPase (NKA),

3. $K^+$-$Cl^-$-cotransporter (KCC),

4. $Na^+/Ca^{2+}$-exchanger (NCX), which is described in the section **Synaptic dynamics**;

5. and Excitatory Amino Acid Transporter (EAAT), which is described in the section **Synaptic dynamics**.

Parameters corresponding to all processes in the neuronal compartment can be found in Table 4.

**Voltage-gated currents and leak channels.** The Goldman-Hodgkin-Katz (GHK) currents are solutions to the Nernst-Planck equations that describe the electrodiffusive flux of ions across a permeable membrane when we assume membrane homogeneity and instantaneous

**Table 3. Common model parameters along with sources.** Units are presented in the same manner as they are implemented in the Python code. All adjusted parameters are in the same order of magnitude as their original counterparts.

| Constant | Value | Description |
|---|---|---|
| $C_n$, $C_a$ | 20 pF | Membrane capacitance [11] |
| F | 96485.333 [C mol$^{-1}$] | Faraday's constant |
| R | 8314.4598 [C(mV)(mol K)$^{-1}$] | Universal gas constant |
| T | 310K | Room temperature [11] |
| $W_{ps}$ | $10^{-3}$ [1000μm$^3$] | Fixed presynaptic terminal volume [59] |
| $W_c$ | $10^{-3}$ [1000μm$^3$] | Fixed synaptic cleft volume (Empirical, chosen to be same as $W_{ps}$) |
| $W_{pap}$ | $10^{-3}$ [1000μm$^3$] | Fixed perisynaptic astrocyte process volume (Empirical, chosen to be same as $W_{ps}$) |

**Table 4. Model parameters for the neuronal compartment, along with sources.** Units are presented in the same manner as they are implemented in the Python code. All adjusted parameters are in the same order of magnitude as their original counterparts.

| Constant | Value | Description |
|---|---|---|
| $P_G^{Na^+,n}$ | $8 \times 10^{-4}$ [1000 $\mu m^3 (ms)^{-1}$] | Voltage-gated $Na^+$ channel permeability [11] |
| $P_G^{K^+,n}$ | $4 \times 10^{-4}$ [1000 $\mu m^3 (ms)^{-1}$] | Voltage-gated $K^+$ channel permeability [11] |
| $P_G^{Cl^-,n}$ | $1.95 \times 10^{-5}$ [1000 $\mu m^3 (ms)^{-1}$] | Voltage-gated $Cl^-$ channel permeability [11] |
| $P_G^{Ca^{2+},n}$ | $1.5 \times 10^{-5}$ [1000 $\mu m^3 (ms)^{-1}$] | Voltage-gated $Ca^{2+}$ channel permeability [60] |
| $P_{NKA}^n$ | 86.4 [pA] | Maximal NKA current (Empirically scaled, of same magnitude as in [11]) |
| $\alpha_{NKA}^{Na^+}$ | 13 [mM] | NKA: Half-saturation concentration for intracellular $Na^+$(Empirical, adjusted from [61]) |
| $\alpha_{NKA}^{K^+}$ | 0.2 [mM] | NKA: Half-saturation concentration for extracellular $K^+$(Empirical, adjusted form [61]) |
| $P_{KCl}^n$ | $1.3 \times 10^{-6}$ [fmol (ms mV)$^{-1}$] | KCl cotransporter strength [11] |
| $P_{NCX}^n$ | 10.8 [pA] | NCX exchanger: scaling factor, identical to astrocyte |
| $\alpha_{NCX}^{Na^+}$ | 87.5 [mM] | NCX exchanger: half saturation concentration for $Na^+$ [61] |
| $\alpha_{NCX}^{Ca^{2+}}$ | 1.38 [mM] | NCX exchanger: half saturation concentration for $Ca^{2+}$ [61] |
| $\eta_{NCX}$ | 0.35 [dimensionless] | NCX exchanger: position of the energy barrier that controls voltage dependence of NCX current [61] |
| $k_{NCX}$ | 0.1 [dimensionless] | NCX exchanger saturation factor at very negative potentials [61] |
| $P_{EAAT}$ | $10^{-6}$ [fmol (ms mV)$^{-1}$] | Neuronal EAAT cotransporter strength, chosen to be of the same magnitude as KCl and NKCC1 cotransporters |
| $\alpha_{H^+}^n$ | 0.66 [dimensionless] | Ratio of extracellular to intracellular proton concentration |
| $L_{H_2O}^n$ | $2 \times 10^{-14}$ [1000 $\mu m^3$ mPa$^{-1}$ ms$^{-1}$] | Neuronal membrane water permeability [11] |

and independent movement of ions. We use the GHK currents to model voltage-gated and leak currents [62]. The gating variables are from [37] and are similar to those in the Hodgkin-Huxley model. The currents are as follows,

$$I_G^{Na^+,n} = P_G^{Na^+,n} m^3 h \, \mathrm{GHK}(V_n, [Na^+]_n, [Na^+]_e),$$
$$I_G^{K^+,n} = P_G^{K^+,n} n^4 \, \mathrm{GHK}(V_n, [K^+]_n, [K^+]_e),$$

(13)

where $m$, $h$ and $n$ are Hodgkin-Huxley gating variables. The function $\mathrm{GHK}(V_y, [X]_y, [X]_e)$ is given by,

$$\mathrm{GHK}(V_y, [X]_y, [X]_e) = \frac{F^2 V_y}{z_X^2 RT} \frac{[X]_y - [X]_e \exp\left(-\frac{FV_y}{z_X RT}\right)}{1 - \exp\left(-\frac{FV_y}{z_X RT}\right)}.$$

(14)

The dynamical equations for $q$ are given by Eq (1) where $q = \{m, h, n\}$. The voltage-dependent expressions $\alpha_q$ and $\beta_q$ are given by,

$$\alpha_m = \frac{0.32(V + 52)}{1 - \exp\left(-(V + 52)/4\right)}, \quad \beta_m = \frac{0.28(V + 25)}{\exp\left((V + 25/5)\right) - 1},$$
$$\alpha_h = 0.128 \exp\left(-(V + 53)/18\right), \quad \beta_h = \frac{4}{1 + \exp\left(-(V + 30)/5\right)},$$
$$\alpha_n = \frac{0.016(V + 35)}{1 - \exp\left(-(V + 35)/5\right)}, \quad \beta_n = 0.25 \exp\left(-\frac{V + 50}{40}\right).$$

(15)

Note that this is different from the subscript $n$, which refers to the neuronal somatic compartment. They have the same expressions as in [37]. For the $Cl^-$ gated current, we adopt the

choice from [11],

$$I_G^{\text{Cl}^-,n} = \frac{P_G^{\text{Cl}^-,n}}{1 + \exp\left(-\frac{V_n+10}{10}\right)} \ \text{GHK}(V_n, [\text{Cl}^-]_n, [\text{Cl}^-]_e). \tag{16}$$

All the leak currents are modelled as GHK currents

$$I_L^{X,n} = P_L^{X,n} \ \text{GHK}(V_n, [X]_n, [X]_e). \tag{17}$$

**Active transport across neuronal membrane: Na$^+$/K$^+$-ATPase (NKA).**   The NKA exchanges three Na$^+$ for two K$^+$ by consuming one molecule of adenosine triphosphate (ATP). It is modelled as a function of intracellular Na$^+$ and extracellular K$^+$ as in [61] by the following flux,

$$I_{NKA}^n(t) = \left(\frac{I_{\text{NKA}}^{\max}(t)}{100}\right) P_{\text{NKA}}^{\text{scale}} f_{\text{NKA}} = \left(\frac{I_{\text{NKA}}^{\max}(t)}{100}\right) P_{\text{NKA}}^{\text{scale}} g_{\text{NKA}} \times$$
$$\frac{[\text{Na}^+]_n^{1.5}}{[\text{Na}^+]_n^{1.5} + (\alpha_{NKA}^n)^{1.5}} \frac{[\text{K}^+]_e}{[\text{K}^+]_e + (\beta_{NKA}^n)}, \tag{18}$$

where

$$g_{\text{NKA}} = \left(1 + 0.1245 \cdot \exp\left(-0.1\frac{FV_n}{RT}\right) + 0.0365 \cdot \sigma \cdot \exp\left(-\frac{FV}{RT}\right)\right), \tag{19}$$

and

$$\sigma = \frac{1}{7} \cdot \left(\exp\left(\frac{[\text{Na}^+]_e}{67.3}\right) - 1\right), \tag{20}$$

where $P_{NKA}^n$ is the NKA permeability or the pump strength. $P_{\text{NKA}}^{\text{scale}}$ is the scaling factor which we vary in simulations to scale the strength of NKA and the function $I_{NKA}^{\max}(t)$ is used to simulate energy derivation for some time period. They are also explained in the section **Simulations and model calibration**. The corresponding Na$^+$ and K$^+$ currents are given by

$$\begin{aligned} I_{\text{NKA}}^{\text{Na}^+,n} &= 3I_{\text{NKA}}^n(t), \\ I_{\text{NKA}}^{\text{K}^+,n} &= -2I_{\text{NKA}}^n(t). \end{aligned} \tag{21}$$

**Secondary active transport across neuronal membrane: K$^+$-Cl$^-$-cotransporter.**   The K$^+$-Cl$^-$-cotransporter (KCC) is a symporter that allows one Cl$^-$ ion and K$^+$ to leave the neuron, along its concentration gradient. It is the main extruder for Cl$^-$ ions in the neuron, thereby providing a counterforce to the gated Cl$^-$ channel, which mediates a massive influx of Cl$^-$ after Na$^+$ loading in neurons [63–67]. We model the flux as the difference of the K$^+$ and Cl$^-$ Nernst potentials as in [40],

$$J_{\text{KCl}} = \frac{RT}{F} \ln\left(\frac{[\text{K}^+]_e[\text{Cl}^-]_e}{[\text{K}^+]_n[\text{Cl}^-]_n}\right). \tag{22}$$

The corresponding K$^+$ and Cl$^-$ currents are given by

$$\begin{aligned} I_{\text{KCl}}^{\text{K}^+,n} &= F J_{\text{KCl}}^n, \\ I_{\text{KCl}}^{\text{Cl}^-,n} &= F J_{\text{KCl}}^n. \end{aligned} \tag{23}$$

## Astrocyte soma

Astrocytes possess a wealth of membrane ion channels and transporters, which allow them to detect, respond and modulate neuronal activity. Major tasks fulfilled by astrocytes at glutamatergic synapses are the regulation of extracellular $K^+$ homeostasis and the re-uptake of synaptically-released glutamate [69]. In this section, we describe the incorporated relevant ion channels/cotransporters that govern astrocyte dynamics during physiological conditions and in response to metabolic stress. We use the following currents/fluxes to describe astrocyte somatic dynamics:

1. Kir4.1 channel,

2. $Na^+/K^+$-ATPase (NKA),

3. $Na^+$-$K^+$-$2Cl^-$-cotransporter (NKCC1),

4. $Na^+/Ca^{2+}$-exchanger (NCX), which is described in the section **Synaptic dynamics**;

5. and Excitatory Amino Acid Transporter (EAAT), which is described in the section **Synaptic dynamics**.

Table 5 lists all parameters corresponding to astrocyte fluxes/currents.

**Kir4.1 channel.** The weakly inwardly rectifying $K^+$ channel Kir4.1 is highly expressed in astrocytes and maintains the resting membrane potential [70, 71], close to the $K^+$ reversal potential. We choose the model from [68], as it allows the current to vanish at the Gibbs-Donnan condition. This property is not present in recently published models on Kir4.1, such as [72]. The current is given by,

$$I_{Kir}^{K^+,a} = P_{Kir} m_\infty \frac{[K^+]_e}{[K^+]_e + 13} \left(V_a - E_{K^+}^a\right), \tag{24}$$

**Table 5. Model parameters for the astrocyte compartment, along with sources.** Units are presented in the same manner as they are implemented in the Python code.

| Constant | Value | Description |
|---|---|---|
| $P_{\text{Kir}}^a$ | 0.286102 [nS] | Kir4.1 conductance, taken from [68] after multiplying with baseline surface area. |
| $P_{NKCC1}^a$ | $7.3215 \times 10^{-7}$ [fmol (ms mV)$^{-1}$] | NKCC1 cotrasporter strength (taken from [40] after multipying with baseline surface area) |
| $P_{EAAT}^a$ | $2 \times 10^{-5}$ [fmol (ms mV)$^{-1}$] | Astrocyte EAAT cotransporter strength, chosen to keep astrocyte to neuronal baseline EAAT current ratio at 9:1. |
| $P_{NKA}^a$ | 86.4 [pA] | Maximal NKA current (Empirically scaled to fit experimental data from [8], of same magnitude as in [11]) |
| $\alpha_{NKA}^{Na^+}$ | 13 [mM] | NKA: Half-saturation concentration for intracellular $Na^+$ [61] |
| $\alpha_{NKA}^{K^+}$ | 0.2 [mM] | NKA: Half-saturation concentration for extracellular $K^+$ [61] |
| $P_{NCX}^a$ | 5.7 [pA] | NCX exchanger: Maximal NCX current, chosen in accordance with [41] as 1/15 of maximal NKA current. |
| $\alpha_{NCX}^{Na^+}$ | 87.5 [mM] | NCX exchanger: half saturation concentration for $Na^+$ [61] |
| $\alpha_{NCX}^{Ca^{2+}}$ | 1.38 [mM] | NCX exchanger: half saturation concentration for $Ca^{2+}$ [61] |
| $\eta_{NCX}$ | 0.35 [dimensionless] | NCX exchanger: position of the energy barrier that controls voltage dependence of NCX current [61] |
| $k_{NCX}$ | 0.1 [dimensionless] | NCX exchanger saturation factor at very negative potentials [61] |
| $\alpha_{H^+}^a$ | 0.66 [dimensionless] | Ratio of extracellular to intracellular proton concentration |
| $L_{H_2O}^a$ | $2 \times 10^{-14}$ [1000 μm$^3$ mPa$^{-1}$ ms$^{-1}$] | Astrocyte membrane water permeability [11] |

with

$$m_\infty = \left( 2 + \exp\left( 1.62 \frac{F}{RT} (V_a - E_{K^+}^a) \right) \right)^{-1}, \tag{25}$$

where $E_{K^+}^a$ is the K$^+$ reversal potential in the astrocyte,

$$E_{K^+}^a = \frac{RT}{F} \log \frac{[K^+]_e}{[K^+]_a}. \tag{26}$$

**Active transport: Na$^+$/K$^+$-ATPase.** The NKA in the astrocyte follows the exact same model as that in the neuron, as in [40]. The NKA current in the astrocyte is given by,

$$I_{NKA}^a = \left( \frac{I_{NKA}^{\max}(t)}{100} \right) P_{NKA}^{\text{scale}} f_{NKA} = \left( \frac{I_{NKA}^{\max}(t)}{100} \right) P_{NKA}^{\text{scale}} g_{NKA} \times$$
$$\frac{[Na^+]_a^{1.5}}{[Na^+]_a^{1.5} + (\alpha_{NKA}^a)^{1.5}} \frac{[K^+]_e}{[K^+]_e + (\beta_{NKA}^a)}, \tag{27}$$

where

$$g_{NKA} = \left( 1 + 0.1245 \cdot \exp\left( -0.1 \frac{F V_a}{RT} \right) + 0.0365 \cdot \sigma \cdot \exp\left( -\frac{F V_a}{RT} \right) \right), \tag{28}$$

and

$$\sigma = \frac{1}{7} \cdot \left( \exp\left( \frac{[Na^+]_e}{67.3} \right) - 1 \right), \tag{29}$$

where $P_{NKA}^a$ is the astrocyte NKA pump strength. Thus the corresponding Na$^+$ and K$^+$ currents are given by

$$I_{NKA}^{Na^+,a} = 3 I_{NKA}^a(t),$$
$$I_{NKA}^{K^+,a} = -2 I_{NKA}^a(t). \tag{30}$$

**Na$^+$-K$^+$-2Cl$^-$-cotransporter (NKCC1).** The astrocyte K$^+$ removal mechanism is complemented by the inward Na$^+$-K$^+$-2Cl$^-$-cotransporter (NKCC1), which is highly expressed in astrocytes [69]. NKCC1 is a symporter and transports one Na$^+$, one K$^+$ and two Cl$^-$ into the astrocyte. It is activated by high extracellular K$^+$ and plays a major role in astrocyte swelling [73, 74]. Astrocyte Cl$^-$ regulation also crucially depends on NKCC1 [75, 76]. The flux is proportional to the difference of Nernst potentials of the respective ions as is done in [40] and is given by

$$J_{NKCC1}^a = P_{NKCC1}^a \frac{RT}{F} \log\left( \frac{[Na^+]_e}{[Na^+]_a} \frac{[K^+]_e}{[K^+]_a} \left( \frac{[Cl^-]_e}{[Cl^-]_a} \right)^2 \right). \tag{31}$$

Thus the corresponding Na$^+$, K$^+$ and Cl$^-$ currents are given by

$$I_{NKCC1}^{Na^+,a} = -F J_{NKCC1}^a,$$
$$I_{NKCC1}^{K^+,a} = -F J_{NKCC1}^a,$$
$$I_{NKCC1}^{Cl^-,a} = 2F J_{NKCC1}^a. \tag{32}$$

**Leak currents.** So far, we only have an inward NKCC1 flux to model movement of Cl$^-$ in the astrocyte. We approximate the remaining fluxes as passive electrodiffusive currents via the

Goldman-Hodgkin-Katz formula for ion currents,

$$I_X^{L,a} = P_X^{L,a} \ \text{GHK}(V_a, [X]_a, [X]_e),$$

(33)

where the formula for $GHK(\cdot, \cdot, \cdot)$ is given by Eq (14), and $X = \{\text{Na}^+, \text{K}^+, \text{Cl}^-\}$.

## Synaptic dynamics

This section describes the dynamics of $\text{Ca}^{2+}$ and glutamate in the synaptic cleft and their coupling to the dynamics of $\text{Na}^+$, $\text{K}^+$ and $\text{Cl}^-$ via the transporters NCX and EAAT. We assume that the volumes of these synaptic compartments remain small and fixed during the first few hours of metabolic stress. Note that, as previously introduced in Eqs (11) and (12), we assume that all of their ions are confined to the 'synaptic compartments' only, i.e., the presynaptic terminal, synaptic cleft and astrocyte process. We now describe the relevant channels/cotransporters acting in the synaptic compartments and the mechanism of glutamate recycling in the cleft.

**Glutamate transport (EAAT).** The re-uptake of synaptically released glutamate is mediated by high-affinity, $\text{Na}^+$-dependent glutamate transporters (EAATs). EAATs are expressed by both presynaptic terminals and astrocytes, with astrocytes mediating about 90% of glutamate uptake in the CNS [14]. The cotransporter protein carries one glutamate molecule, three $\text{Na}^+$ and one $\text{H}^+$ into the cells in exchange for one $\text{K}^+$. The transport yields a net double positive charge influx. We model EAAT in the same way the KCC and NKCC1 cotransporters are modelled. Thus, the EAAT current is written as,

$$J_{EAAT}^i = P_{EAAT}^i \frac{\text{RT}}{\text{F}} \ln \left( \frac{[\text{Na}^+]_e^3 \, [\text{K}^+]_i \, [\text{H}^+]_c \, [\text{Glu}]_c}{[\text{Na}^+]_i^3 \, [\text{K}^+]_e \, [\text{H}^+]_i \, [\text{Glu}]_i} \right).$$

(34)

The corresponding ion currents are

$$
\begin{aligned}
I_{EAAT}^{\text{Na}^+,i} &= -3\text{F}J_{EAAT}^i, \\
I_{EAAT}^{\text{K}^+,i} &= \text{F}J_{EAAT}^i, \\
I_{EAAT}^{\text{Glu},i} &= \text{F}J_{EAAT}^i.
\end{aligned}
$$

(35)

In our model, we do not model the dynamics of protons, but we keep the ratio $\frac{[\text{H}^+]_c}{[\text{H}^+]_a}$ constant [77]. The constant is chosen from modelling work done in [39], where EAAT forms a part of the biophysical description of ion homeostasis at the postsynaptic cradle.

**$\text{Na}^+/\text{Ca}^{2+}$-exchanger (NCX).** The $\text{Na}^+/\text{Ca}^{2+}$-exchanger NCX allows $\text{Na}^+$ to flow along its concentration gradient into the neuron/astrocyte in exchange for $\text{Ca}^{2+}$. It is expressed across various cell types, including neurons and astrocytes. Thus, three $\text{Na}^+$ are imported in exchange for one $\text{Ca}^{2+}$, yielding a net positive charge to the compartment. The notable aspect about NCX is that it reverses when $[\text{Na}^+]$ increases in the respective compartment [8].

We follow the model from [61], which describes the NCX current by

$$
\begin{aligned}
I_{NCX}^i = P_{NCX}^i &\left( \frac{[\text{Na}^+]_e^3}{\alpha_{\text{Na}^+}^3 + [\text{Na}^+]_e^3} \right) \left( \frac{[\text{Ca}^{2+}]_c}{\alpha_{\text{Ca}^{2+}} + [\text{Ca}^{2+}]_c} \right) \times \\
&\frac{\frac{[\text{Na}^+]_i^3}{[\text{Na}^+]_e^3} \exp\left( \frac{\eta \text{F} V_i}{\text{RT}} \right) - \frac{[\text{Ca}^{2+}]_i}{[\text{Ca}^{2+}]_c} \exp\left( \frac{(\eta-1)\text{F} V_i}{\text{RT}} \right)}{1 + k_{NCX} \exp\left( \frac{(\eta-1)\text{F} V_i}{\text{RT}} \right)}.
\end{aligned}
$$

(36)

The corresponding $Na^+$ and $Ca^{2+}$ currents are given by

$$I_{NCX}^{Na^+,i} = 3I_{NCX}^i,$$
$$I_{NCX}^{Ca^{2+},i} = -I_{NCX}^i. \tag{37}$$

**Vesicular recycling.** In response to action potentials that reach the presynaptic terminal and consequent $Ca^{2+}$ elevations, glutamate is released into the synaptic cleft. This process involves packing of the neurotransmitter into synaptic vesicles, which fuse with the presynaptic membrane following formation of the SNARE complex [78]. The packing of glutamate into vesicles by vesicular glutamate transporters (VGLUTs) depends on $[Cl^-]$ and on a proton gradient across the vesicular membrane, which is mediated by an ATP-dependent proton pump. It was suggested that VGLUT expression declines with age, although these ideas still remain inconclusive [79]. In this work, we assume that vesicular packing and recycling is not directly energy-dependent. This allows us to see what happens during partial ED when $Na^+/K^+$-ATPase is affected, but glutamate continues to be efficiently packed into vesicles.

To model vesicular recycling, we combine models from [42] and [43], see Fig 2. The model proposed by Walter et al. [43] describes the sequential slow-fast mechanism of packing glutamate into vesicles, depending on $Ca^{2+}$ elevations. The sequence models the pathway of glutamate from a large storage pool (called depot) to the irreversible fused state, which is when glutamate is released into the synaptic cleft by the interaction of vesicles with SNARE proteins lined up on the presynaptic membrane. The cycle is then completed by plugging in the model by Tsodyks and Markram [42] which models glutamate recycling back into the depot. The equations for the inactive (I) and fused (F) state are then adjusted to include the uptake of glutamate via EAAT and electrodiffusive leak currents. This is done by removing the linear term for recruitment of glutamate from the fused state back into the inactive state, and replacing it with terms for EAAT and leak dynamics. The dynamical equations for the various states of glutamate are given by,

$$
\begin{cases}
\dfrac{d}{dt}N_I = \dfrac{d}{dt}N_{Glu}^n = -\dfrac{1}{\tau_{rec}}N_I N_D + \dfrac{1}{F}(I_{EAAT}^{Glu,n} + I_L^{Glu,n}), \\[2mm]
\dfrac{d}{dt}N_D = \dfrac{1}{\tau_{rec}}N_I N_D - k_1 N_D + k_{-1}N_N, \\[2mm]
\dfrac{d}{dt}N_N = k_1 N_D - (k_{-1} + k_2)N_N + k_{-2}N_R, \\[2mm]
\dfrac{d}{dt}N_R = k_2 N_N - (k_{-2} + 3k_3[Ca^{2+}]_n)N_R + k_{-3}N_{R_1}, \\[2mm]
\dfrac{d}{dt}N_{R_1} = 3k_3[Ca^{2+}]_n N_R - (k_{-3} + 2k_3[Ca^{2+}]_n)N_{R_1} + 2k_{-3}N_{R_2}, \\[2mm]
\dfrac{d}{dt}N_{R_2} = 2k_3[Ca^{2+}]_n N_{R_1} - (2k_{-3} + k_3[Ca^{2+}]_n)N_{R_2} + 3k_{-3}N_{R_3}, \\[2mm]
\dfrac{d}{dt}N_{R_3} = k_3[Ca^{2+}]_n N_{R_2} - (3k_{-3} + k_4)N_{R_3}.
\end{cases} \tag{Mod.1}
$$

The coefficient $k_1$ is taken from [43]

$$k_1 = k_{1,max}\frac{[Ca^{2+}]_n}{[Ca^{2+}]_n + K_M} \tag{38}$$

where $K_M$ is the half saturation $Ca^{2+}$ concentration in the presynaptic terminal to recruit

vesicles from the depot into the non-releasable pool. The coefficients $k_2$ and $k_{-2}$ are taken from [43]

$$k_2(\text{Ca}^{2+}) = k_{20} + g(\text{Ca}^{2+})k_{2\text{cat}},$$
$$k_{-2}(\text{Ca}^{2+}) = k_{-20} + g(\text{Ca}^{2+})k_{-2\text{cat}}, \tag{39}$$

where the probability for a $\text{Ca}^{2+}$-bound catalyst $g(\text{Ca}^{2+})$ is given by [43]

$$g(\text{Ca}^{2+}) = \frac{[\text{Ca}^{2+}]}{[\text{Ca}^{2+}] + K_{\text{Dv}}}. \tag{40}$$

From the molar amounts of the various states of glutamate, we can define glutamate concentrations in the neuron and cleft. Thus,

$$[\text{Glu}]_n = [\text{Glu}]_{ps} = \frac{1}{W_{PreSyn}} N^n_{\text{Glu}} = \frac{1}{W_{PreSyn}}(N_I) \tag{41}$$

and

$$[\text{Glu}]_c = \frac{1}{W_c} N^c_{\text{Glu}} = \frac{1}{W_c} N_F. \tag{42}$$

All parameters corresponding to vesicular recycling can be found in Table 6. The expression for $N_F$ is derived from conservation laws, see section **Conservation laws**.

## Volume dynamics

The exact channels for water movement between the extracellular space, neuron and astrocytes is are still debated today [80, 81]. In our work, we assume it to depend linearly on the osmotic pressure gradient across the membrane. We follow the model from [11] to model the volume

**Table 6. Model parameters for glutamate recycling, along with sources.** Units are presented in the same manner as they are implemented in the Python code.

| Constant | Value | Description |
|---|---|---|
| $k_1^{\max}$ | 1 [ms$^{-1}$] | Maximum forward reaction rate (Empirical) |
| $K_M$ | $2.3 \times 10^{-3}$ [mM] | $\text{Ca}^{2+}$ half-saturation concentration for forward reaction rate (Depot to Non Releasable Pool) [43] |
| $K_{D_v}$ | $1 \times 10^{-4}$ [mM] | Half-saturation concentration for forward reaction rate (Non releasable pool to readily releasable pool) [43] |
| $k_{20}$ | $2.1 \times 10^{-5}$ [ms$^{-1}$] | Uncatalysed forward reaction rate [43] |
| $k_{2_{\text{cat}}}$ | $2 \times 10^{-2}$ [ms$^{-1}$] | Catalysed forward reaction rate [43] |
| $k_{-20}$ | $1.7 \times 10^{-5}$ [ms$^{-1}$] | Uncatalysed backward reaction rate [43] |
| $k_{-1}$ | $5 \times 10^{-5}$ [ms$^{-1}$] | Backward reaction rate [43] |
| $k_3$ | 4.4 [(mM ms)$^{-1}$] | Forward reaction rate [43] |
| $k_{-3}$ | $5.6 \times 10^{-2}$ [ms$^{-1}$] | Backward reaction rate [43] |
| $k_4$ | 1.45 [ms$^{-1}$] | Fusion rate [43] |
| $\tau_{\text{rec}}$ | 30 [ms (fmol)$^{-1}$] | Vesicle fusion factor (Empirical, adapted from [43]) |

compartmental volume $W_i$ as

$$\frac{d}{dt} W_i = L^i_{H_2O} \Delta \pi_i, \tag{Mod.2}$$

where $\Delta \pi_i$ is the osmotic pressure gradient given by,

$$\Delta \pi_i = \text{RT} \sum_X ([X]_i - [X]_e), \tag{43}$$

for $X, Y \in \{\text{Na}^+, \text{K}^+, \text{Cl}^-\}$ and $i \in \{n, a\}$. From Eq (1), we then obtain the relation

$$\lambda_i = L^i_{H_2O} \text{RT}. \tag{44}$$

## Model equations

The dynamics of individual ion amounts based on the described ion currents is given by

$$
\begin{cases}
\dfrac{d}{dt} N^n_{\text{Na}^+} = -\dfrac{1}{\text{F}} (I^{\text{Na}^+,n}_{\text{G}} + I^{\text{Na}^+,n}_{\text{NKA}} + I^{\text{Na}^+,n}_{\text{EAAT}} + I^{\text{Na}^+,n}_{\text{NCX}} + I^{\text{Na}^+,n}_{\text{L}}) + \dfrac{1}{\text{F}} I_{\text{stim}}(t), \\[2mm]
\dfrac{d}{dt} N^n_{\text{K}^+} = -\dfrac{1}{\text{F}} (I^{\text{K}^+,n}_{\text{G}} + I^{\text{K}^+,n}_{\text{NKA}} + I^{\text{K}^+,n}_{\text{EAAT}} + I^{\text{K}^+,n}_{\text{KCl}} + I^{\text{K}^+,n}_{\text{L}}), \\[2mm]
\dfrac{d}{dt} N^n_{\text{Cl}^-} = \dfrac{1}{\text{F}} (I^{\text{Cl}^-,n}_{\text{G}} + I^{\text{Cl}^-,n}_{\text{KCl}} + I^{\text{Cl}^-,n}_{\text{L}}), \\[2mm]
\dfrac{d}{dt} N^n_{\text{Ca}^{2+}} = -\dfrac{1}{2\text{F}} (I^{\text{Ca}^{2+},n}_{\text{G}} + I^{\text{Ca}^{2+},n}_{\text{NCX}} + I^{\text{Ca}^{2+},n}_{\text{L}}), \\[2mm]
\dfrac{d}{dt} N^n_{\text{Glu}} = \dfrac{1}{\text{F}} (I^{\text{Glu},n}_{\text{EAAT}} + I^{\text{Glu},n}_{L}), \\[2mm]
\dfrac{d}{dt} N^a_{\text{Na}^+} = -\dfrac{1}{\text{F}} (I^{\text{Na}^+,a}_{\text{NKCC1}} + I^{\text{Na}^+,a}_{\text{NKA}} + I^{\text{Na}^+,a}_{\text{EAAT}} + I^{\text{Na}^+,a}_{\text{NCX}} + I^{\text{Na}^+,a}_{\text{L}}), \\[2mm]
\dfrac{d}{dt} N^a_{\text{K}^+} = -\dfrac{1}{\text{F}} (I^{\text{K}^+,a}_{\text{NKCC1}} + I^{\text{K}^+,a}_{\text{NKA}} + I^{\text{K}^+,a}_{\text{EAAT}} + I^{\text{K}^+,a}_{\text{Kir}} + I^{\text{K}^+,a}_{\text{L}}), \\[2mm]
\dfrac{d}{dt} N^a_{\text{Cl}^-} = \dfrac{1}{\text{F}} (I^{\text{Cl}^-,a}_{\text{NKCC1}} + I^{\text{Cl}^-,a}_{L}), \\[2mm]
\dfrac{d}{dt} N^a_{\text{Ca}^{2+}} = -\dfrac{1}{2\text{F}} (I^{\text{Ca}^{2+},a}_{\text{NCX}} + I^{\text{Ca}^{2+},a}_{\text{L}}), \\[2mm]
\dfrac{d}{dt} N^a_{\text{Glu}} = \dfrac{1}{\text{F}} (I^{\text{Glu},a}_{\text{EAAT}} + I^{\text{Glu},a}_{L}),
\end{cases}
\tag{Mod.3}
$$

where $I_{\text{stim}}(t)$ is a square-wave current used to stimulate the neuron when we perform neuronal stimulation experiments, such as in Fig 4. The Eqs (Mod.1)–(Mod.3) together give the model. The initial values used, are shown in Table 7. However, we have not described extracellular dynamics yet. These are obtained directly from conservation laws, which we describe in the following section.

## Conservation laws

The model has constant total volume $W_{\text{tot}}$, i.e.

$$\sum_i W_i = W_{tot}. \tag{Cons.1}$$

**Table 7. Initial values for the various states in the model.** These values correspond to 'baseline' conditions, and are used to estimate unknown parameters. Units are presented in the same manner as they are implemented in the Python code.

| Constant | Value | Description |
|---|---|---|
| $V_i^0$ | -65.5 [mV] | Neuronal membrane potential at rest. |
| $V_g^0$ | -80 [mV] | Astrocyte membrane potential at rest. |
| $[\text{Na}^+]_n^0$ | 13 [mM] | Taken from [8]. |
| $[\text{K}^+]_n^0$ | 145 [mM] | Taken from [11]. |
| $[\text{Cl}^-]_n^0$ | 7 [mM] | Taken from [11]. |
| $[\text{Ca}^{2+}]_n^0$ | $1 \times 10^{-4}$ [mM] | Rounded off from 73 nM [82]. |
| $[\text{Glu}]_n^0$ | 2.2385 [mM] | Free glutamate in the cytoplasm is about 2 mM [83]. See section Estimating parameters. |
| $[\text{Na}^+]_a^0$ | 13 [mM] | Obtained from experimental traces in [8]. |
| $[\text{K}^+]_a^0$ | 80 [mM] | Obtained by setting $\text{K}^+$ reversal potential to $\sim$-85 mV (here it is -87.7 mV). |
| $[\text{Cl}^-]_a^0$ | 35 [mM] | Obtained from Bergmann glia data in [84]. |
| $[\text{Ca}^{2+}]_a^0$ | $1 \times 10^{-4}$ [mM] | Chosen to be the same as in the presynaptic terminal (Empirical). |
| $[\text{Glu}]_a^0$ | 2 [mM] | Chosen to be the same as in the presynaptic terminal (Empirical). |
| $m_0$ | $1.33135 \times 10^{-2}$ | $\text{Na}^+$ activation gating variable. Estimated by setting right-hand side of the third equation of Eq (1) to zero at resting conditions. |
| $h_0$ | 0.987298 | $\text{Na}^+$ inactivation gating variable. Estimated by setting right-hand side of the third equation of Eq (1) to zero at resting conditions. |
| $n_0$ | $2.96946 \times 10^{-3}$ | $\text{K}^+$ activation gating variable. Estimated by setting right-hand side of the third equation of Eq (1) to zero at resting conditions. |
| $N_I^0$ | $2.238 \times 10^{-3}$ [fmol] | Baseline molar amount of free glutamate in the presynaptic terminal. Obtained from the relation $N_I^0 = [\text{Glu}]_n^0 \times W_{\text{PreSyn}}$. |
| $N_D^0$ | $4.04605 \times 10^{-7}$ [fmol] | Baseline molar amount of vesicular glutamate in the depot of the presynaptic terminal. See section Estimating parameters. |
| $N_N^0$ | $3.36567 \times 10^{-4}$ | Baseline molar amount of non releasable vesicular glutamate in the presynaptic terminal. See section Estimating parameters. |
| $N_R^0$ | $4.14849 \times 10^{-4}$ | Baseline molar amount of readily releasable vesicular glutamate (not yet binded to $\text{Ca}^{2+}$) in the presynaptic terminal. See section Estimating parameters. |
| $N_{R_1}^0$ | $9.778061 \times 10^{-6}$ | Baseline molar amount of readily releasable vesicular glutamate (binded to one $\text{Ca}^{2+}$ ion) in the presynaptic terminal. See section Estimating parameters. |
| $N_{R_2}^0$ | $7.655809 \times 10^{-8}$ | Baseline molar amount of readily releasable vesicular glutamate (binded to two $\text{Ca}^{2+}$ ions) in the presynaptic terminal. |
| $N_{R_3}^0$ | $2.08192593 \times 10^{-11}$ | Baseline molar amount of readily releasable vesicular glutamate (not yet binded to three $\text{Ca}^{2+}$) in the presynaptic terminal. |
| $W_n^0$ | 2 [1000μm³] | Baseline neuronal soma volume (taken from [11]) |
| $W_a^0$ | 2 [1000μm³] | Baseline astrocyte soma volume (Empirical, chosen to be the same as $W_n^0$) |

As a consequence we get a conservation law for ionic molar amounts giving

$$\sum_i N_X^i = C_X, \tag{Cons.2}$$

where the sum is over all compartments, for each ion *X*. As the net charge in the system must be zero, we have, at all times

$$\sum_X z_X C_X + \sum_{Y,i} z_Y N_Y^i = 0, \tag{Cons.3}$$

where *Y* contains the impermeable cations and anions. The Eqs (Cons.1)–(Cons.3) give us the three conserved quantities. As a consequence, we can now describe extracellular dynamics from

$$W_e = W_{tot} - W_n - W_a,$$

$$[X]_e = \frac{1}{W_e}\left(C_X - \sum_{i \neq e} N_X^i\right). \tag{45}$$

## Estimating parameters from conservation laws

In order to maintain physiological resting states, we incorporate impermeable ions in the system. Biophysically these correspond to large proteins that are unable to move across the cell membrane. We calculate them directly from conservation equations. At rest, the right-hand side of Eq (Mod.2) must be equal to zero. From this, we get two rest conditions,

$$
\begin{cases}
\sum_X \left([X]_n - [X]_e\right) = 0, \\
\sum_X \left([X]_a - [X]_e\right) = 0,
\end{cases}
\tag{Rest.1}
$$

where $X$ accounts for all ions (including impermeable ones) in the system. At baseline conditions, Eqs (10) and (Cons.3) provide three more rest conditions,

$$
\begin{cases}
\sum_X z_X C_X + \sum_{Y,i} z_Y N_Y^i = 0, \\
V_n^0 = \dfrac{\mathrm{F}}{\mathrm{C_i}} \sum_X z_X [X]_n^0 W_n^0, \\
V_a^0 = \dfrac{\mathrm{F}}{\mathrm{C_i}} \sum_X z_X [X]_a^0 W_a^0,
\end{cases}
\tag{Rest.2}
$$

**Table 8. Parameters estimated from baseline conditions.** Units are presented in the same manner as they are implemented in the Python code. See section 'Estimating parameters' for derivation.

| Constant | Value | Description |
|---|---|---|
| $P_L^{Na^+,n}$ | $1.706 \times 10^{-6}$ [1000μm³(ms)⁻¹] | Neuronal Na⁺ leak channel permeability. |
| $P_L^{K^+,n}$ | $1.771 \times 10^{-5}$ [1000μm³(ms)⁻¹] | Neuronal K⁺ leak channel permeability. |
| $P_L^{Cl^-,n}$ | $2.494 \times 10^{-6}$ [1000μm³(ms)⁻¹] | Neuronal Cl⁻ leak channel permeability. |
| $P_L^{Ca^{2+},n}$ | $1.649 \times 10^{-11}$ [1000μm³(ms)⁻¹] | Neuronal Ca²⁺ leak channel permeability. |
| $P_L^{Glu,n}$ | $3.662 \times 10^{-6}$ [1000μm³(ms)⁻¹] | Neuronal Glu leak channel permeability. |
| $P_L^{Na^+,a}$ | $1.054 \times 10^{-7}$ [1000μm³(ms)⁻¹] | Astrocytic Na⁺ leak channel permeability. |
| $P_L^{K^+,a}$ | $7.877 \times 10^{-5}$ [1000μm³(ms)⁻¹] | Astrocytic K⁺ leak channel permeability. |
| $P_L^{Cl^-,a}$ | $4.388 \times 10^{-7}$ [1000μm³(ms)⁻¹] | Astrocytic Cl⁻ leak channel permeability. |
| $P_L^{Ca^{2+},a}$ | $3.022 \times 10^{-10}$ [1000μm³(ms)⁻¹] | Astrocytic Ca²⁺ leak channel permeability. |
| $P_L^{Glu,a}$ | $2.891 \times 10^{-5}$ [1000μm³(ms)⁻¹] | Astrocytic Glu leak channel permeability. |
| Volumes and amounts of ions (obtained after setting $\alpha_e = 20\%$) | | |
| $W_e^0$ | 0.925 [1000μm³] | Baseline extracellular volume. |
| $N_{A^-}^n$ | 302.0105 [fmol] | Amount of impermeant anions in the neuronal soma. |
| $N_{B^+}^e$ | 2.790 [fmol] | Amount of impermeant cations in the extracellular space. |
| $N_{A^-}^e$ | 21.264 [fmol] | Amount of impermeant anions in the extracellular space. |
| $N_{B^+}^a$ | 110.497 [fmol] | Amount of impermeant cations in the astrocyte soma. |
| $N_{A^-}^a$ | 209.111 [fmol] | Amount of impermeant anions in the astrocyte soma. |
| $C_{Na^+}$ | 188.7 [fmol] | Total amount of Na⁺ions in the system. |
| $C_{K^+}$ | 428.775 [fmol] | Total amount of K⁺ions in the system. |
| $C_{Cl^-}$ | 198.375 [fmol] | Total amount of Cl⁻ions in the system. |
| $C_{Ca^{2+}}$ | $1.8 \times 10^{-3}$ [fmol] | Total amount of Ca²⁺ions in the system. |
| $C_{Glu}$ | $5 \times 10^{-3}$ [fmol] | Total amount of Gluions in the system. |
| $W_{tot}$ | 2.925 [1000μm³] | Total volume of the system. |

where $Y = \{A^-, B^+\}$ are the impermeable ions. Using Eqs (Rest.1) and (Rest.2), we compute the constants $N_{A^-}^n$, $N_{A^-}^e$, $N_{B^+}^e$, $N_{A^-}^a$ and $N_{B^+}^a$.

We assume that total glutamate in the presynaptic terminal amounts to 2 mM. Thus,

$$\sum_Z N_Z^0 = 2 \times W_{ps}, \quad \text{where } Z \in \{I, N, D, R, R1, R2, R3\}. \tag{Rest.3}$$

Thus, from Eq Rest.3 and by setting the right hand side of Eq 2 to 0 at baseline conditions, we can compute all initial conditions corresponding to the various glutamate stages.

The leak permeabilities $P_L^{X,i}$ are computed by setting the dynamical equations of $N_X^i$ from Eq Mod.3 to zero at rest conditions. Note that for glutamate and $Ca^{2+}$ dynamics, we assume constant volumes of the presynaptic terminal, synaptic cleft and perisynaptic astrocytes processes.

The various parameters estimated in this section are laid out in Table 8.

## Python implementation

The code is implemented in Python and is available publicly at github.com/mkalia94/TripartiteSynapse. The simulations were made with the CVode solver, implemented in the Python package assimulo.

## Supporting information

**S1 Fig. Extension of Fig 4.** Here, we further plot $Cl^-$ and $Ca^{2+}$ concentration profiles, along with important ion gradients. (B) shows that astrocytes are critical for maintaining ion homeostasis when $Na^+/K^+$-ATPase (NKA) is fully functional. Here, we plot neuronal (blue), astrocyte (orange) and extracellular (green) traces against time for several quantities. The initial extracellular volume ratio $\alpha_e$ = 20%. Shaded red area corresponds to periods during which ion transport across the astrocytic plasma membrane is blocked. Neurons are subjected to a 25 pA square wave input, as indicated by the black trace. The burst contains 475 action potentials. The green extracellular calcium trace has an offset of 1.8 mM.
(EPS)

**S2 Fig. Extension of Fig 5.** Here, we further plot $Cl^-$ and $Ca^{2+}$ concentration profiles, along with important ion gradients. We plot neuronal (blue), astrocyte (orange) and extracellular (green) traces against time. The initial extracellular volume ratio $\alpha_e$ = 80% and minimal energy available $P_{min}$ = 50%. Shaded grey areas correspond to the period where $Na^+/K^+$-ATPase (NKA) activity is gradually reduced to $P_{min}$ and restored to baseline after 5 minutes (left panel) or 15 minutes (right). This temporal profile is indicated in the upper panel.
(EPS)

**S3 Fig. Extension of Fig 7B.** We introduce pharmacological blockers post ED by blocking various ion channels/cotransporters to look for potential for recovery from the ED-induced pathological state. In all of the cases presented, the pathological state remains stable even after channel blockade. We plot neuronal (blue) and astrocyte (orange) membrane potentials against time. The initial extracellular volume ratio $\alpha_e$ = 80% and minimal energy available $P_{min}$ = 0%. Shaded grey areas correspond to the period where $Na^+/K^+$-ATPase (NKA) activity is gradually reduced to $P_{min}$ and restored to baseline after 15 minutes. The shaded light green area corresponds to channel blockade.
(EPS)

## Acknowledgments

We thank Dr. Christoph Fahlke, FZ Jülich, Germany, for helpful discussions.

## Author Contributions

**Conceptualization:** Manu Kalia, Hil G. E. Meijer, Stephan A. van Gils, Michel J. A. M. van Putten, Christine R. Rose.

**Formal analysis:** Manu Kalia.

**Funding acquisition:** Stephan A. van Gils, Christine R. Rose.

**Investigation:** Manu Kalia.

**Methodology:** Manu Kalia.

**Software:** Manu Kalia.

**Supervision:** Hil G. E. Meijer.

**Validation:** Manu Kalia, Hil G. E. Meijer, Stephan A. van Gils, Michel J. A. M. van Putten, Christine R. Rose.

**Visualization:** Manu Kalia.

**Writing – original draft:** Manu Kalia, Hil G. E. Meijer, Michel J. A. M. van Putten, Christine R. Rose.

**Writing – review & editing:** Manu Kalia, Hil G. E. Meijer, Stephan A. van Gils, Michel J. A. M. van Putten, Christine R. Rose.

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
