## [Decision Letter · Decision Letter 0]

12 Nov 2020

Dear Mr. Kalia,

Thank you very much for submitting your manuscript "Ion dynamics of the energy-deprived tripartite synapse" for consideration at PLOS Computational Biology.

As with all papers reviewed by the journal, your manuscript was reviewed by members of the editorial board and by several independent reviewers. In light of the reviews (below this email), we would like to invite the resubmission of a significantly-revised version that takes into account the reviewers' comments.

The reviewers made a large number of important points regarding the writing style and figure presentation, and reported serious issues regarding potassium and sodium dynamics in the model. Please make sure to take into account all those points.

We thank you for participating in the reproducibility pilot and congratulate you for successfully passing the reproducibility test.

We cannot make any decision about publication until we have seen the revised manuscript and your response to the reviewers' comments. Your revised manuscript is also likely to be sent to reviewers for further evaluation.

Sincerely,

Hugues Berry

Associate Editor

PLOS Computational Biology

Daniele Marinazzo

Deputy Editor

PLOS Computational Biology

Reviewer's Responses to Questions

**Comments to the Authors:**

Reviewer #1: Reproducibility report has been uploaded as an attachment.

Reviewer #2: The manuscript by Kalia and colleagues titled "Ion dynamics of the energy-deprived tripartite synapse" develops a model of the tripartite synapse in physiology and in periods of energetic stress. The manuscript is extremely well written and very clear. It represents a commendable effort by the authors to develop an almost exhaustive model of ion transport at the tripartite synapse, and is a perfect fit for this journal in this reviewer's opinion. The authors proceed to compare and analyse the dynamics of the system when facing the challenge of energy deprivation of various durations and severity. The work is a very nice computational biology study with broad relevance in neuroscience. This reviewer's enthusiasm is however tempered slightly by some issues that could be important for the validity of the results.

Major

* Both the time and concentration scales in Fig. 3B are different, which makes it difficult to evaluate how well the model reproduces experimental data after calibration. Please adjust these.

* There seems to be an issue with the potassium and sodium dynamics in the model, which does not reproduce or capture experimental results accurately. This is apparent in Fig. 3B again, where the undershoot after a period of energy deprivation present in experimental data is not captured by the model. Similarly, we would expect sodium in astrocytes to go up when the synapse is activated. The model predicts it goes down instead. The authors cite their own work but that paper actually says: "Astrocyte sodium concentration increased by up to 8.5 mM, which could be followed by an undershoot below baseline." and the text of the present manuscript says: "In our simulation, astrocyte Na+ decreases in response to neuronal burst firing, which is also observed experimentally under conditions of recurrent network activity and activation of astrocyte NKA by increases in extracellular K+ [41]. Notably, when Na+-dependent glutamate uptake or influx through NMDA receptors override the NKA-induced export of Na+, a net increase in astrocyte Na+ is induced [41]." There seems to be an important discrepancy here between what the model does when the synapses is activated and what experiments observe. This reviewer thinks that this is due to how glutamate recapture is implemented in the model. Figure 2 suggests for instance that this only happens in neurons. The authors do say in line 540 that 90% of the reuptake happens in astrocytes however, which is probably correct experimentally and roughly matches the distribution of these transporters as observed in the Barres transcriptome. I believe the authors should make sure that the 90% glutamate uptake is true overall and not only at baseline (see Table 5). That might solve this issue.

* It is unclear to this reviewer how the volumes of the cells are modelled. Eq 1 in page 4 does not seem compatible with Eq. Mod. 2 in page 23. Please clarify.

Minor

* Some figure references are missing around line 213.

* Figures 7 and 8 appear to be swapped, or their legends are.

* The authors might want to check the model of brain energy metabolism published by the Magistretti group in 2015 (PLOS Comput Biol) for the sake of completeness.

Reviewer #3: Dear Authors,

It was very interesting to read your manuscript "Ion dynamics of the energy-deprived tripartite synapse". I have some comments which should be addressed.

Introduction:

Page 2, line 21: Since you model the volume changes, it would be helpful to explain the mechanism of cell swelling in 1 or 2 sentences.

P 2, l 40: Please include some review articles here: e.g., doi.org/10.3389/fncom.2018.00014, doi.org/10.1016/j.brainresbull.2017.01.027, book doi.org/10.1007/978-3-030-00817-8

Model Description:

P 5, equation 3: Describe C_W and C_X in the text.

P 5, Figure 2: Please extend the figure caption. In the figure itself, what does the bracket mean in "R1: Vesicle bound -1 Ca2+)" (same for R_2)?

P 5, l 134: For the reading flow, it is helpful if you shortly describe f^i_NKA at this point and then refer to Materials and Methods (as done).

P 6, l 139: In my opinion, the sentence is ambiguous. It is not clear if the whole recording is 2 minutes long or the ischemic part.

P 6, Figure 3: Do you have an explanation for why you have a sharper peak in the neuronal Na+ and why your modeled peaks are at the end of the ischemia period while they are earlier in the experiments?

For a modeler, it would be interesting to know how you have done the fitting process.

Results:

P 7, l 159: I suggest to exchange" Experiments" by" Simulations" to avoid confusion.

P 8, l 188-191: Please refer to the corresponding figure again.

P 8, Figure 5 caption: What means" gradually reduced" and" restored back"? Please specify if this was done over a time period or instantaneously.

P 9, l 219: The figure number is confusing.

P 9, l 242: I suggest to change the order of alpha_e = 20% and alpha_e= 80 % in this line since the text afterward and the figure start with alpha_e= 80 %.

Figure 7: Please be aware that you had swapped Figures 7 and 9 when you uploaded them.

Figures 6 and 8: I suggest to put Figure 6-A and 8-A together in one. It seems more logical to me.

Figure 6-B: Please use the same x- and y-axes for the subfigures. It would be easier to compare them.

P 11, l 305-306: Please remove the sentence "In Fig. 6-B.3,4, P^scale_NKA is set to 2." You have mentioned it already in line 297.

P 12, l 348: I suggest to add the not shown data to the supplement.

Discussion:

I found the following interesting review article: "Molecular mechanisms of K+ clearance and extracellular space shrinkage—Glia cells as the stars " (https://doi.org/10.1002/glia.23824). Maybe you find more evidence in the article that supports your findings.

Conclusions:

P 14, l 459ff.: Please be more specific on your findings.

Material and methods:

P 15, equation 10: I could not find the description of z_X in this section.

Supplementary Figure S1: What does the "+1.8" mean in the right y-axis label of the Ca2+ concentration? Please specify in the caption.

Reviewer #4: Kalia and co-workers present a detailed biophysical model of ion homeostasis at cortical synapses, including astrocytic signaling. Astrocyte-regulated synapses, aka tripartite synapses, represent 50-80% of cortical synapses, but their physiology remains an open matter of active investigation. This study fills in this line, focusing on neuron-glial-based ion homeostasis in setting ischemic conditions, both in the healthy and the aged brain. This is undoubtedly an exciting study, which aims at tracking down biophysical pathways for the complex topic of ion (dys)homeostasis in the brain, and the fact that it takes into account glial signaling makes it one of the few studies in this sense currently available in the literature – and arguably one of the most detailed ones. Accordingly, I deem the manuscript of interest for PLoS Computational Biology. However, I am afraid I cannot recommend it for publication in its current version due to some severe pitfalls. The writing style is generally poor, and manuscript and figures' organization can be vastly improved. Accordingly, I am suggesting significant revisions before re-considering the manuscript for publication. Please refer to the attached PDF for suggestions for editing. Thank you for your consideration.

**Have all data underlying the figures and results presented in the manuscript been provided?**

Reviewer #1: Yes

Reviewer #2: Yes

Reviewer #3: Yes

Reviewer #4: **No: **Figure 9 is missing. Material in provided Figure 6B, 7B and 9 appears not to be in an orderly fashion.

PLOS authors have the option to publish the peer review history of their article (what does this mean?). If published, this will include your full peer review and any attached files.

Reviewer #1: **Yes: **Anand K. Rampadarath

Reviewer #2: No

Reviewer #3: **Yes: **Kerstin Lenk

Reviewer #4: No
---

## [Decision Letter · Decision Letter 1]

28 Apr 2021

Dear Mr. Kalia,

We are pleased to inform you that your manuscript 'Ion dynamics at the energy-deprived tripartite synapse' has been provisionally accepted for publication in PLOS Computational Biology.

Best regards,

Hugues Berry

Associate Editor

PLOS Computational Biology

Daniele Marinazzo

Deputy Editor

PLOS Computational Biology

Reviewer's Responses to Questions

**Comments to the Authors:**

Reviewer #3: Dear Authors,

Thank you for considering my suggestions and answering my questions!

Reviewer #4: Dear authors,

I appreciate your edits and although I believe, you should have implemented all my original suggestions, especially the one that you rejected, I understand your motivations. Therefore I am promoting the paper to publication. Nice work.

Cheers.

**Have all data underlying the figures and results presented in the manuscript been provided?**

Reviewer #3: Yes

Reviewer #4: Yes

PLOS authors have the option to publish the peer review history of their article (what does this mean?). If published, this will include your full peer review and any attached files.

Reviewer #3: **Yes: **Kerstin Lenk

Reviewer #4: No

---

## [Editor Report · Acceptance letter]

9 Jun 2021

PCOMPBIOL-D-20-01345R1 

Ion dynamics at the energy-deprived tripartite synapse

Dear Dr Kalia,

I am pleased to inform you that your manuscript has been formally accepted for publication in PLOS Computational Biology. Your manuscript is now with our production department and you will be notified of the publication date in due course.

With kind regards,

Olena Szabo
